# Antigen discrimination by T cells relies on size-constrained microvillar contact

Edward Jenkins [1,2,4], Markus Körbel [3,4], Caitlin O'Brien-Ball [1,2], James McColl[3], Kevin Y. Chen [3], Mateusz Kotowski [1,2], Jane Humphrey [3], Anna H. Lippert[3], Heather Brouwer[1,2], Ana Mafalda Santos[1,2], Steven F. Lee[3], Simon J. Davis [1,2] ✉ & David Klenerman [3] ✉

T cells use finger-like protrusions called 'microvilli' to interrogate their targets, but why they do so is unknown. To form contacts, T cells must overcome the highly charged, barrier-like layer of large molecules forming a target cell's glycocalyx. Here, T cells are observed to use microvilli to breach a model glycocalyx barrier, forming numerous small (<0.5 μm diameter) contacts each of which is stabilized by the small adhesive protein CD2 expressed by the T cell, and excludes large proteins including CD45, allowing sensitive, antigen dependent TCR signaling. In the absence of the glycocalyx or when microvillar contact-size is increased by enhancing CD2 expression, strong signaling occurs that is no longer antigen dependent. Our observations suggest that, modulated by the opposing effects of the target cell glycocalyx and small adhesive proteins, the use of microvilli equips T cells with the ability to effect discriminatory receptor signaling.

The initiation of adaptive immune responses relies on T cells forming contacts with other cells. This enables, especially, T-cell receptors (TCRs) to engage 'foreign' peptide fragments complexed with major histocompatibility complex proteins (pMHC) on target and antigen-presenting cells (APCs), leading to the initiation of signaling and eventual clearance of pathogens and tumors. From the earliest days of the application of scanning electron microscopy, lymphocytes were distinguished by the presence of numerous finger-like surface projections called "microvilli", in contrast, e.g., to monocytes that mainly displayed ruffled membranes and ridge-like protrusions[1]. It is now known that microvilli are used by T cells to efficiently scan target cells for antigen. Using lattice light-sheet imaging, Cai et al. showed that 98% of the T cell/APC interface could be visited by microvilli within one minute[2]. But why T cells use microvilli instead of larger structures to scan targets is largely unknown. One explanation is that the use of microvilli represents the preferred energetic solution for penetrating the glycocalyx of apposing cells and detecting the relatively short ligands of the TCR[2]. A second, non-exclusive possibility, supported by in silico simulations based on a "dwell-time" model of TCR signaling[3], is

that individual contacts must be small in order for ligand discrimination to be possible.

The glycocalyx is a negatively charged, dense array of glyco-conjugates and/or glycoproteins that extends 50–500 nm from most cell membranes, comprising an important barrier to cell contact and adhesion[4–8]. In the case of nucleated cells of hemopoietic origin, the two key glycoprotein elements of the glycocalyx are CD43 and CD45[9–12], with the former also frequently expressed by (non-hemopoietic cell-derived) tumors[13,14]. In the absence of active processes, the CD43/CD45-based glycocalyx will likely prevent interactions between T cells and APCs at distances shorter than 50–100 nm. Accordingly, CD43 has been shown to antagonize cell contact[15–17] and, whereas the impact of CD45 on cell adhesion is less well-studied, it has been estimated that the spontaneous local exclusion of CD45 from a 100 nm diameter region of cell surface, allowing passive microvillar contact, would take $10^9$ s (31.7 years)[18]. This suggests that there will be a requirement for T cells to physically exclude CD45 on opposing surfaces to establish contacts[19]. Since T cells spend just 1–5 min searching their targets for antigen[20], active processes will need to drive close

[1]Radcliffe Department of Medicine, John Radcliffe Hospital, University of Oxford, Oxford OX3 9DS, UK. [2]Medical Research Council Human Immunology Unit, John Radcliffe Hospital, University of Oxford, Oxford OX3 9DS, UK. [3]Yusuf Hamied Department of Chemistry, University of Cambridge, Cambridge CB2 1EW, UK. [4]These authors contributed equally: Edward Jenkins, Markus Körbel. ✉e-mail: simon.davis@imm.ox.ac.uk; dk10012@cam.ac.uk

membrane approximation long enough for adhesive proteins to stabilize T cell/target contact, allowing efficient pMHC scanning and engagement by the TCR.

The two most important adhesive proteins on the surface of T cells that are likely to mediate contact are CD2 and LFA-1[21]. CD2 is a relatively small (7.5 nm) single-pass transmembrane protein composed of immunoglobulin superfamily domains that binds a similarly small, evolutionarily related protein, CD58. In contrast, LFA-1 is a much larger (20–25 nm) heterodimer, that binds intercellular adhesion molecules (ICAMs) 1-5 and junctional adhesion molecules 1-2, with the strongest-binding and best-characterized ligand being ICAM-1 (19 nm)[22–24]. CD58 and ICAM-1 are ubiquitously expressed on hemopoietic and non-hemopoietic cells in humans, underscoring the importance of CD2 and LFA-1 to T-cell function[14–16]. These proteins enhance T-cell activation through their adhesive and costimulatory properties and are considered important therapeutic targets and risk factors for a variety of pathologies[25–34]. In particular, the CD2/CD58 pair has recently been identified as having an important function in the etiology of cancer[35–38].

Here, we show that microvilli are required to penetrate a model glycocalyx, forming numerous small, uniform contacts that are stabilized by CD2/ligand interactions and facilitate ligand engagement by TCRs. Importantly, altering the size of the individual contacts allowed us to also show that TCR discrimination requires each contact to be size-limited. These data suggest that the use of microvilli equips T cells with a capacity for discriminatory receptor signaling.

## Results

### A supported lipid-bilayer mimic of the APC surface

T-cell interactions are typically studied with supported lipid-bilayers (SLBs) presenting pMHC and ICAM-1 extracellular domains (ECDs) only (SLB1s; Fig. 1a)[39]. To study the likely interplay between microvilli, a glycocalyx, and adhesion proteins, we created more complex SLBs that mimic the APC cell surface by presenting physiological densities of 'null' pMHC, agonist pMHC, ICAM-1, CD58, and a model glycocalyx comprising the two major glycoprotein elements of the glycocalyx formed by APCs, i.e., CD43 and CD45 (for the present experiments, CD45RABC), utilizing histidine tags to anchor the ECDs of each protein to the SLB (see Supplementary Fig. 1a–e for quantification of the proteins versus human monocyte-derived dendritic cells, and SLB configurations). The complex of the '9V' variant of the cancer/testis NY-ESO-1 peptide (157–165; SLLMWITQV) and HLA-A2 (pMHC[9V]), which binds the 1G4 TCR[40] with a solution $K_D$ of 7.2 μM[41], was used as the agonistic ligand. HLA-A2 complexed with the non-cognate gp100 peptide (YLEPGPVTV) served as the null ligand (pMHC[null]). To study T-cell engagement in a ligand-dependent manner, we generated a human CD4⁻CD8αβ⁺ 1G4 TCR-expressing Jurkat T cell line[42], transduced with additional LFA-1 to match the surface phenotype of primary CD8⁺ effector T cells (Supplementary Fig. 1f), and a genetically encoded calcium indicator (GECI) to monitor intracellular calcium release[43] as a marker of TCR triggering/early T-cell activation. GECI levels were kept low to prevent it behaving as a calcium sink[44]. We refer to this as the 'J8-GECI' cell line.

We first examined the impact of CD58 and the model glycocalyx on pMHC sensing. On SLB1s presenting pMHC[null] with and without increasing densities of pMHC[9V] (0.01–100 molecules/μm²; see Supplementary Fig. 1g for pMHC[9V] density measurements), the fraction of J8-GECI cells that produced a calcium response increased from ~10% to ~85%, taking between ~1000 s to ~100 s to reach a 50% response level (Fig. 1a). Adding CD58 (SLB1 + CD58) caused a striking increase in the extent and tempo of the response, at the cost of a large reduction in pMHC specificity, since ≥60% of cells responded to pMHC[null] and pMHC[9V] at all densities within 200 s (Fig. 1b). The increase in responsiveness was dependent on the TCR but not CD2 signaling (considered below). Adding the model glycocalyx to the SLB1 substantially reduced calcium responses at all pMHC[9V] levels (by >30%) and introduced

delays in the response (up to 2–3-fold; Fig. 1c). These effects are likely due to reduced contact formation, reflecting the barrier effects of the glycocalyx[5], due in part to its heavy glycosylation (with, e.g., sialic acid; Supplementary Fig. 1h). Adding both CD58 and the glycocalyx balanced sensitivity and specificity, returning the response curve to near SLB1 levels, albeit with an enhanced response to pMHC[null] (25% vs 10%), and an approximately 2-fold faster response at low pMHC[9V] levels (Fig. 1d). We refer to this configuration, i.e., the presentation of CD58 and a glycocalyx alongside pMHC and ICAM-1, as the 'second generation' SLB (SLB2). Overall, responses to SLB1 and SLB2 bilayers were comparable; however, the SLB2 allowed the interplay between a glycocalyx and adhesion proteins to be studied.

We further investigated the countervailing effects of adhesion and the model glycocalyx on responses to pMHC[null] by analyzing, alongside calcium signaling, cell spreading as a measure of activation using interference reflection microscopy (IRM), and synapse formation (Supplementary Fig. 1i)[19,45]. Removal of either glycocalyx element (CD43 or CD45) from SLB2s presenting pMHC[null], had a modest impact on calcium signaling, contact area, and synapse formation (Fig. 1e, f). However, removal of both dramatically increased signaling, cell spreading, and synapse formation, producing responses comparable to those with pMHC[9V]. In contrast, removing CD58, but not ICAM-1, from SLB2s reduced the fraction of cells that responded to pMHC[null]. Strikingly, removing either or both adhesive proteins blocked spreading and synapse formation, confirming the importance of adhesive protein-mediated contact formation in enhancing T-cell responsiveness. Primary T cells responded similarly to the J8-GECI cell line on null- versus agonist-presenting SLB2s in terms of both calcium signaling (Supplementary Fig. 1j) and synapse formation (Supplementary Fig. 1k). These results confirmed the important functions of glycocalyces and adhesive proteins in securing the specificity and sensitivity of early T-cell signaling, respectively. As described in what follows, we explored contact formation on SLB2s and how this affects T-cell antigen recognition and discrimination.

### Overcoming a glycocalyx barrier

T cells are believed to use microvilli to search for pMHC on apposing cells[2,46]. Having shown that the presence of a glycocalyx substantially suppresses early signaling events, we sought to show that microvilli are instrumental in overcoming glycocalyx barriers. Electron microscopy confirmed that the surfaces of J8-GECI cells were populated with microvilli (Supplementary Fig. 2a), that were enriched for L-selectin at their tips (Supplementary Fig. 2b)[46]. We labeled CD43 and CD45 inserted into SLB2s presenting pMHC[9V] at ~100 molecules/μm², and the cell membrane of J8-GECI cells, and then imaged the cells as they formed contacts with the SLBs using total internal reflection fluorescence microscopy (TIRFM). Prior to the cell settling on the surface, we observed small puncta of membrane fluorescence appearing and disappearing, suggestive of active surface sampling by the cell (Supplementary Fig. 2c). The puncta persisted for ~8 s (Supplementary Fig. 2d); similar observations were made previously for primary T cells[47]. Once a cell had settled, small and distinct 'holes' in the fluorescent CD43/45 layer began to form within the area bounded by the membrane fluorescence (Supplementary Movie 1). Given the similarity in (1) the sizes of the membrane puncta prior to contact, (2) the size of the holes subsequently created in the bilayer fluorescence, and (3) the dimensions of the microvillar protrusions of J8-GECI cells (~0.45 μm diameter; Supplementary Fig. 2e), we conclude that, in these experiments, the model glycocalyx is being breached by microvilli, creating the holes in the fluorescence of the SLB2. To analyze protein localization at these sites, we developed a bespoke segmentation and analysis pipeline that utilized gradient-based filtering and hysteresis thresholding to automatically segment the images (see "Methods" for details). Formation of the holes in the

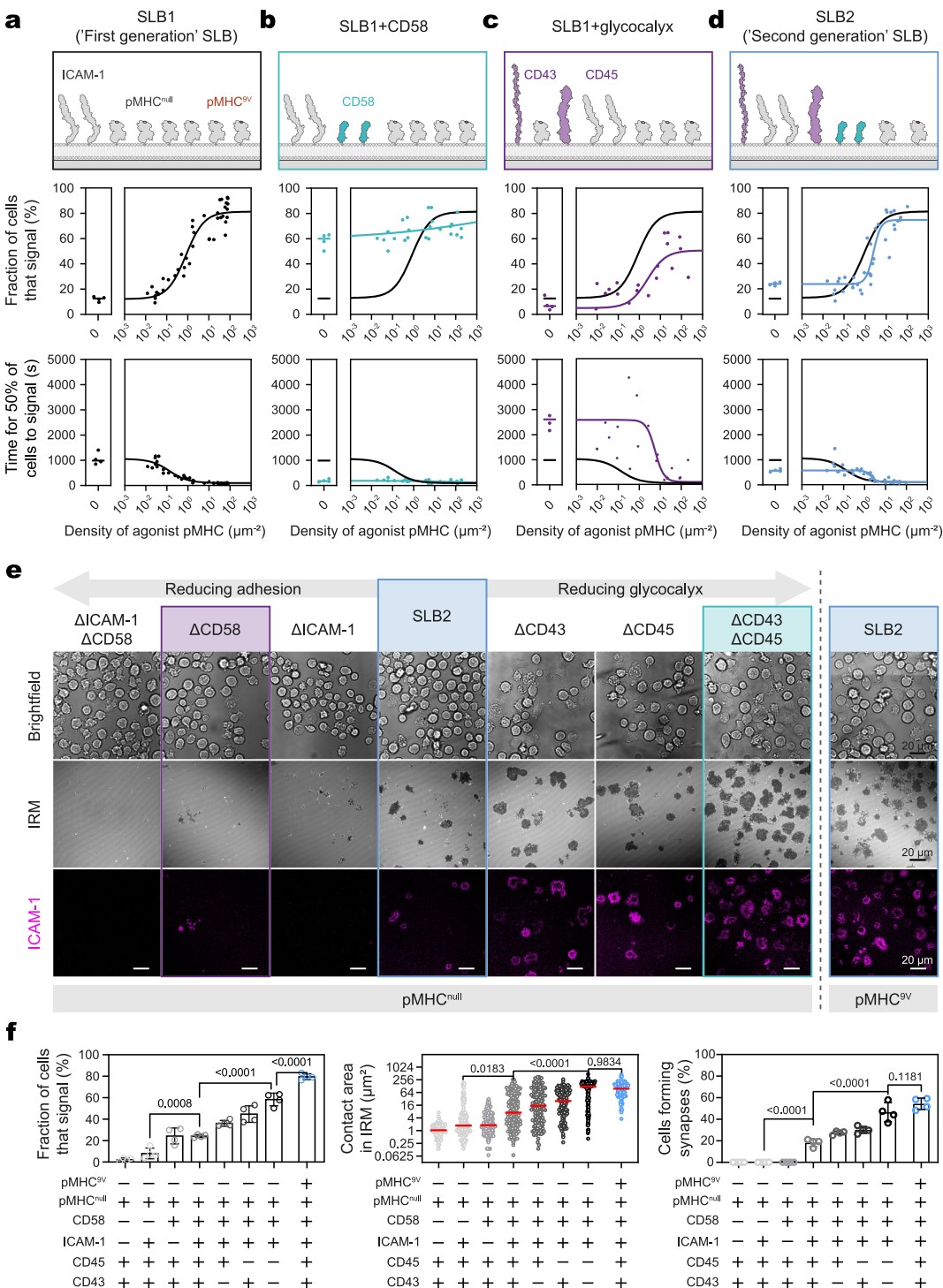

CD43/45 fluorescence correlated with heightened TCR (Supplementary Fig. 2f), pMHC (Supplementary Fig. 2g), and ZAP70 fluorescence (Supplementary Fig. 2h). L-selectin was initially observable but was later excluded (Supplementary Fig. 2i; Supplementary Movie 2). We also confirmed that the holes in the fluorescence were formed by exclusion of the glycocalyx, rather than other photophysical effects, by showing that fluorescent CD43/45 immobilized directly onto a glass surface exhibited uniform fluorescence during cell-contact formation (Supplementary Fig. 2j). We refer to the holes in the fluorescence observed on SLB2s, created by microvilli and marked by the local exclusion of glycocalyx elements, which correspond to regions of TCR engagement and signaling, as 'close contacts'.

To further confirm that microvilli are needed to actively penetrate a glycocalyx barrier and establish close contacts, we used a suite of actin-modifying drugs to alter or 'paralyze' J8-GECI microvillus formation (Fig. 2a, b). Using our segmentation and analysis pipeline, we found that the loss of microvilli, or their paralysis, reduced the number of cells overcoming the glycocalyx barrier to form close contacts (Fig. 2c, d), and slowed contact formation (Fig. 2e; Supplementary Movie 3). Cells unable to form or to mobilize microvilli produced substantially smaller calcium responses on SLB2s (Fig. 2f). The treated cells were nevertheless capable of calcium signaling in the absence of the glycocalyx, albeit with slightly reduced capacity (Fig. 2f). These results confirmed the need for actin remodeling to overcome a

**Fig. 1 | SLB2s balance antigen sensitivity and specificity. a–d** Calcium response curves for J8-GECI cells on the indicated SLBs presenting pMHC$^{null}$ ± pMHC$^{9V}$. A cartoon indicating the SLB compositions is shown (top panel), along with the fraction of cells exhibiting a Ca$^{2+}$ signal (middle panel) and the time taken for 50% of the cells to signal (bottom panel). Each datapoint corresponds to a separate SLB. Data were fitted with a four-point dose-response curve, constrained to a minimum using responses to pMHC$^{null}$. **a** "First generation" SLB ('SLB1'; black); $n = 44$ SLBs with ≥116 cells analyzed per SLB. **b** SLB1 + CD58 (cyan); $n = 28$ SLBs with ≥127 cells analyzed per SLB. **c** SLB1 + glycocalyx (magenta); $n = 23$ SLBs with ≥140 cells analyzed per SLB. **d** "Second generation" SLB ('SLB2'; blue); $n = 42$ SLBs with ≥109 cells analyzed per SLB. **e, f** Spreading, synapse formation, and calcium release for J8-GECI cells interacting with non-adhesive (ΔICAM-1ΔCD58) to highly adhesive (ΔCD43ΔCD45) SLBs. All the SLBs presented pMHC$^{null}$, except for the right-most SLB2, which presented pMHC$^{null}$ plus pMHC$^{9V}$ at ~100 molecules/μm². J8-GECI cells were tracked for calcium signals for 10 min and immediately imaged afterward to allow IRM-based contact area measurement and synapse formation frequency.

**e** Images of cells spreading (dark areas in IRM image) and forming synapses. Colored boxes denote the same SLB composition shown in (**a–d**). Images are representative of J8-GECI cells on $n = 4$ independent SLBs for each SLB composition. **f** Left plot: fraction of cells that exhibit a calcium signal. Shown is the mean (±S.D.) of $n = 4$ independent SLBs with ≥158 cells analyzed per SLB. ΔCD43ΔCD45, ΔCD58, and SLB2s presenting pMHC$^{null}$ correspond to the zero density values in (**b**), (**c**), and (**d**), respectively. **f** Middle plot (log scale): quantification of cell spreading; $n = 115$ (ΔICAM-1ΔCD58), 133 (ΔCD58), 174 (ΔICAM-1), 154 (SLB2 pMHC$^{9V-}$), 132 (ΔCD43), 108 (ΔCD45), 118 (ΔCD43ΔCD45), and 122 SLB2 (SLB2 pMHC$^{9V+}$) cells pooled from four independent SLBs for each SLB composition. The red line indicates median. **f** Right plot: quantification of synapse formation. Shown is the mean (±S.D.) fraction of cells forming synapses from $n = 4$ independent SLBs with ≥17 cells analyzed per SLB. In **f**, means were compared to an SLB2 presenting pMHC$^{null}$ via one-way ANOVA with Tukey correction. Comparison between the ΔCD43ΔCD45 SLB and SLB2 (with pMHC$^{9V}$) was also included. Only statistics for comparisons of interest are shown. Source data are provided in the Source data file.

---

glycocalyx barrier and efficiently detect pMHC, via the formation of microvilli[48].

## The four stages of close-contact formation

To explore close-contact formation and its relationship to pMHC sensing, we used three-color TIRF-based imaging to simultaneously analyze the triggering state (calcium signaling), interaction footprint (cell membrane staining), and close contacts (fluorescent glycocalyx components of the SLB) for cells interacting with SLB2s presenting pMHC$^{null}$ and two levels of pMHC$^{9V}$ ligands (pMHC$^{9V-lo}$, i.e., ~1 molecule/μm² and pMHC$^{9V-hi}$, i.e., ~100 molecules/μm²). Supplementary Movie 1, and Supplementary Movies 4, and 5, show J8-GECI cells interacting with SLB2s presenting both pMHC$^{9V-hi}$ and pMHC$^{null}$, and just pMHC$^{null}$ (signaling cells and non-signaling cells), respectively. For cells interacting with SLB2s presenting pMHC$^{9V-hi}$, we identified four distinct stages of contact: 'searching', 'scanning', 'spreading', and 'synapsing' (Fig. 3a). Searching (stage I) was marked by microvillar tips moving in and out of the evanescent field with no observable close contacts being formed. The scanning stage (stage II) was initiated by close-contact formation, marked by holes in the glycocalyx fluorescence, which, importantly, persisted throughout the following stages. The formation of close contacts presumably allows the TCR to scan for cognate pMHC, which we explore further below. As dynamic actin remodeling was key to efficient contact formation (Fig. 2d), the transition between searching and scanning is likely a stochastic process, dependent on intrinsic pathways governing protrusion dynamics. Calcium signaling marked the onset of spreading (stage III), which led to an increase in both the cell footprint and number of close contacts. Stage III ended with the beginning of the synapsing stage (IV), characterized by contraction of the footprint and centripetal movement of the close contacts.

The transition from searching to scanning occurred independently of the nature of the pMHC (Fig. 3b), whereas the progression to spreading was pMHC sensitive (Fig. 3c). For example, non-signaling cells on pMHC$^{null}$-presenting SLB2s could penetrate the glycocalyx but then exhibited only small increases in cell footprint (Fig. 3c) or close-contact number (Fig. 3d). In contrast, signaling cells on pMHC$^{9V-hi}$- and pMHC$^{9V-lo}$-presenting SLB2s exhibited large increases in cell footprint and number of close contacts formed, although spreading was accelerated on the pMHC$^{9V-hi}$-presenting SLB2s. Cells responding to pMHC$^{null}$ exhibited smaller increases in cell footprint and contact formation than with pMHC$^{9V-lo}$, indicative of there being a relationship between signal strength and spreading[19]. A striking finding was that, whereas the cell footprint and close-contact number each increased with the level of stimulation up to and during the spreading stage (i.e., up to 150 s for cells on pMHC$^{9V-hi}$ SLB2s, and >300 s for other surfaces), the initial close-contacts formed (i.e., during the first 10 s of observation) were

uniformly small (0.2–0.4 μm in diameter), even after calcium signaling (Fig. 3e). During the synapsing stage close-contact size increased, owing to the consolidation of contacts at the center of the cell. We address the importance of close-contact size below.

We also examined close-contact formation for primary CD8$^+$ T cells on SLB2s presenting pMHC$^{null}$ ± ~20 molecules/μm² of a UCHT-1 Fab-HaloTag construct used as a strong agonist (SLB-bound UCHT-1 Fab has been used previously as a 'pan-specific' TCR ligand for human T cells[29,49,50]). Responding primary cells loaded with Fluo-4 exhibited all four stages of contact formation (Supplementary Movie 6; see Supplementary Movie 7 for a non-signaling cell). Although the main features of their interactions with agonist- and non-agonist-presenting SLB2s were broadly similar for primary cells (Fig. 3f–i), compared to J8-GECI cells, primary cells had a smaller cellular footprint (maximum of 127 versus 229 μm²; Fig. 3g), and formed more close contacts (82 versus 65 contacts/cell; Fig. 3h) of comparable size over time (Fig. 3i). Collectively, these results indicate that antigen detection during the early stages of T-cell activation occurs exclusively at small, microvillus-generated close contacts.

## Impact of glycocalyx density on close-contact formation

Although the glycocalyx of the SLB2 was designed to mimic that of an APC, we sought to understand how contact formation is affected by glycocalyx density, which is known to vary[13,14,30]. We therefore increased the density of the SLB2 glycocalyx three-fold, whilst keeping other protein densities constant (e.g., CD58; Supplementary Fig. 2k). This reduced, to some extent, the fraction of J8-GECI cells producing calcium responses for all conditions but had a modest, if any, effect on signaling times (Supplementary Fig. 2l). Increasing the density of the glycocalyx slightly delayed the transition from searching to scanning (22 s vs. 4 s, $p = 0.003$; Supplementary Fig. 2m). For cells that produced calcium signals we observed a reduction in the cellular footprint (maximum 90 versus 175 μm²), the number of close contacts formed (maximum 40 versus 60 contacts), and the extent of exclusion of the glycocalyx at each close contact (a proxy for contact 'tightness'; 27% versus 35%; Supplementary Fig. 2n). Importantly, the areas of the initially formed close contacts (~0.3 μm in diameter) remained constant (Supplementary Fig. 2n), further indicating that close-contact size is set, at least in part, by the dimensions of microvilli as they penetrate glycocalyx barriers. These results suggest that T cells can penetrate glycocalyx barriers of varying density to form close contacts, albeit with differing efficiency.

## Individual microvilli are capable of efficient antigen detection

We observed pMHC (Fig. 3j; Supplementary Movie 8) and ZAP70 (Fig. 3k; Supplementary Movie 9) accumulation at close contacts formed by J8-GECI cells interacting with pMHC$^{9V-hi}$-presenting SLB2s during the scanning stage, prior to calcium release, confirming that

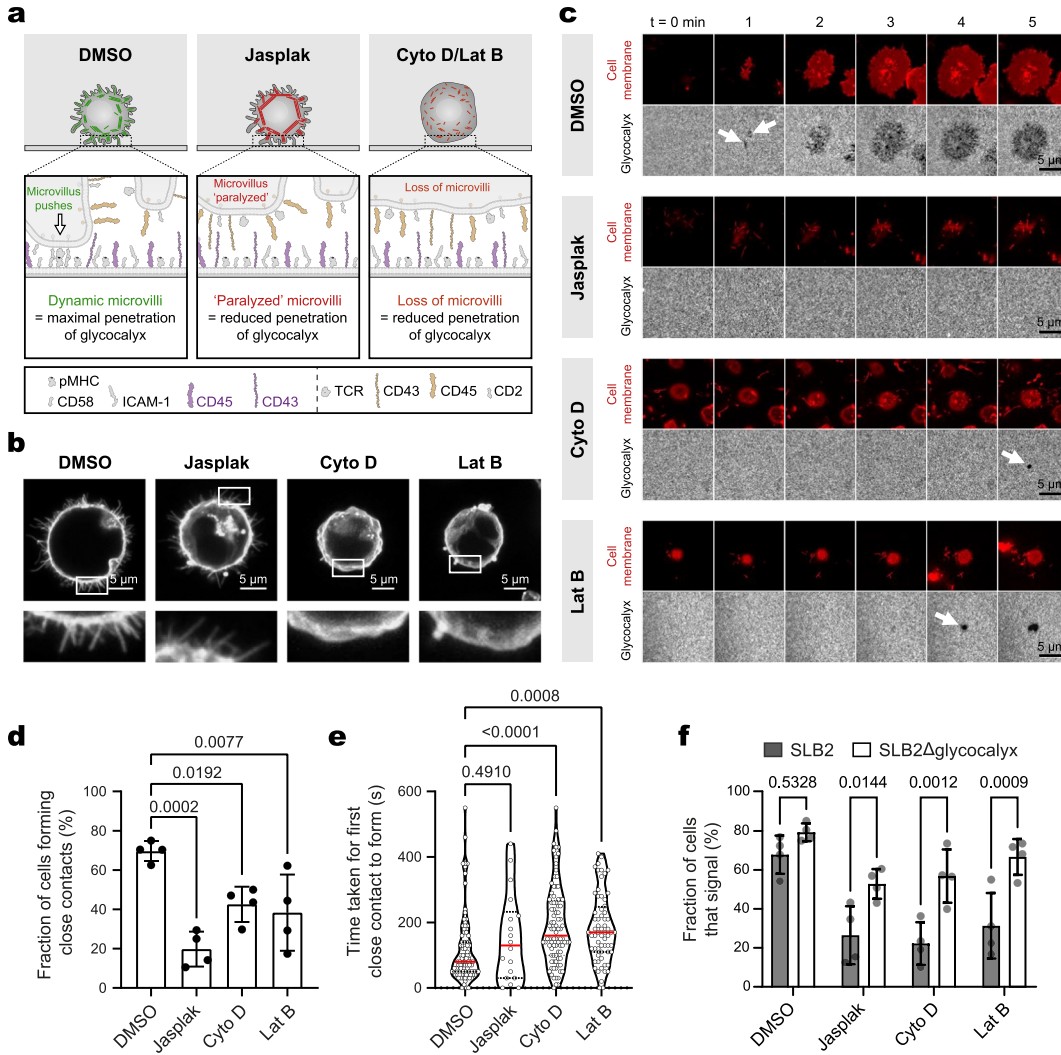

**Fig. 2 | Microvilli allow T cells to overcome a glycocalyx barrier. a** Cartoon depiction of experiments testing whether dynamic actin remodeling of microvilli is needed to establish close-contacts in the presence of a glycocalyx barrier. Jasplakinolide (Jasplak) enhances the nucleation and stabilization of actin filaments, resulting in 'paralysis' of the actin cytoskeleton and microvillar activity. Latrunculin B (Lat B) and cytochalasin D (Cyto D) prevent the formation of actin filaments by sequestering actin monomers and by blocking their recruitment to pre-existing filaments, respectively, resulting in filament breakdown and subsequent loss of membrane topography. Cells in the cartoon are shown interacting with an SLB2. **b** Confocal fluorescence images of fixed J8-GECI cells imaged at the midplane after treatment with DMSO or an actin-modifying drug. A digitally-magnified region (white box) is shown below. Images are representative of J8-GECI cells for $n = 3$ independent experiments. **c** Examples of contact formation in the presence of DMSO or the actin-modifying drugs, for J8-GECI cells interacting with an SLB2 presenting pMHC$^{null}$ plus pMHC$^{9V\text{-}lo}$, i.e., -1 pMHC$^{9V}$ molecule/μm². Close contacts are indicated by black holes in the SLB2 glycocalyx fluorescence (with point of initiation indicated by white arrows). See Supplementary Movie 3. Images are representative of drug-treated J8-GECI cells for $n = 4$ independent experiments. **d** Fraction of cells that formed detectable close contacts. Data are from the same experiment as in (**c**). Shown is the mean (±S.D.) of $n = 4$ independent SLBs with 13–113 cells imaged per SLB. **e** Time taken for cells to form a close contact versus first appearance of membrane fluorescence. Data are from the same experiment as in (**c**). Data were pooled from four independent SLBs with $n = 93$ (DMSO), 88 (Cyto D), 18 (Jasplak), and 64 (Lat B) total contact-forming cells analyzed. The red line indicates the median. In **d** and **e**, means were tested using one-way ANOVA with Dunnett correction using DMSO as the control group. **f** Fraction of DMSO- or actin-modifying drug-treated J8-GECI cells that exhibit calcium release on an SLB2 or SLB2Δglycocalyx (i.e., SLB1 + CD58) presenting pMHC$^{null}$ plus pMHC$^{9V\text{-}lo}$. Shown is the mean (±S.D.) of $n = 4$ independent SLBs with 138-785 cells analyzed per SLB. Two-way ANOVA with Šidák correction was used to compare means between glycocalyx-positive and -negative SLBs for each treatment. Source data are provided in the Source data file.

close contacts are sites of TCR engagement and receptor triggering. Measurements of contact area at calcium release indicated that pMHC sensing was highly efficient, requiring <0.2% of the total cellular surface area (Fig. 3l) shared across fewer than 10 close contacts, irrespective of pMHC$^{9V}$ density (Fig. 3m); primary T cells used slightly more surface area and extra close-contacts, on average (Fig. 3m). In some instances, J8-GECI cells (Fig. 3m; Supplementary Movie 8) and primary cells (Fig. 3m) initiated signaling after forming 1–2 close contacts with the agonist-presenting SLB2. We therefore tested whether single pMHCs engaged at close contacts could mobilize signaling. By detecting bound pMHC on SLB2s presenting very low densities of

pMHC$^{9V}$, measured over the lifetime of the contact, we found that very small numbers of pMHC$^{9V}$ (i.e., 1–2 molecules) were engaged per close contact in cells that signaled, leading to cell spreading (Supplementary Movie 10), whereas none were bound by cells that did not signal (Fig. 3n, o). These results suggest that the use of microvilli allows for highly sensitive antigen detection by T cells.

## CD2, but not LFA-1, stabilizes close-contact formation

Previous work implied that TCR occupancy is required for microvillar contact stabilization[2]. We speculated that, in the presence of a glycocalyx, and in the more physiological setting of low agonist pMHC

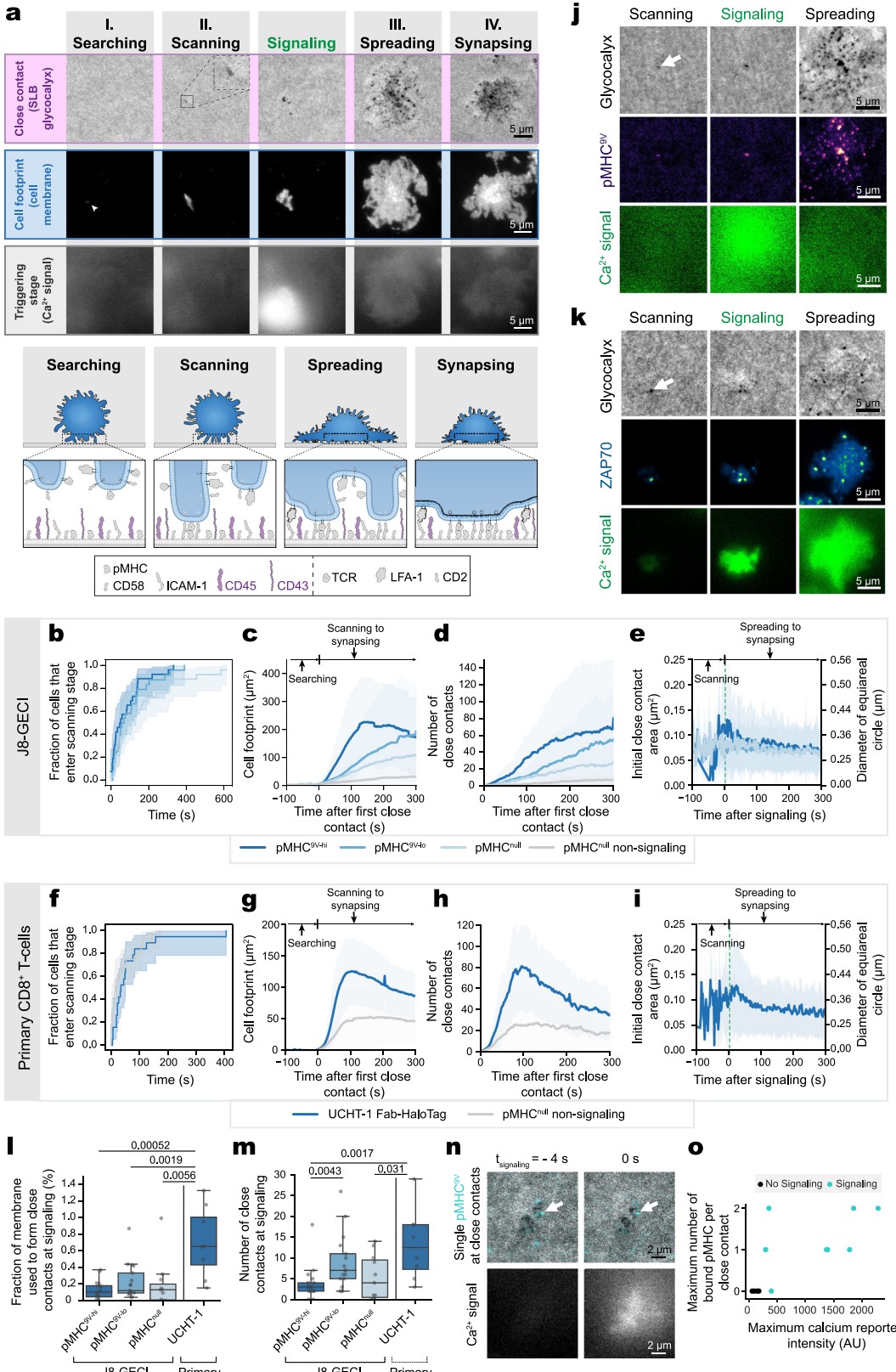

levels, adhesion molecules would be key to stabilizing contacts. To test this, we generated TCR-deficient J8-GECI T cells (TCRKO cells; Supplementary Fig. 3a). The TCRKO cells did not produce calcium responses on OKT3-coated glass or pMHC^null-presenting SLB2s (Supplementary Fig. 3b, c), and neither did they spread on the SLB2s, despite forming contacts detectable using IRM (Supplementary

Fig. 3d). TIRFM confirmed that these cells entered the scanning stage but did not transition to the spreading stage (Supplementary Fig. 3e; Supplementary Movie 11). The close contacts were persistent (Supplementary Fig. 3f) and similar in number and size to those formed by non-signaling TCR-expressing cells (Supplementary Fig. 3g). For these cells, the maximum CD58 fluorescence intensity and track-length for

**Fig. 3 | The four stages of close-contact formation. a** Key stages of close-contact formation. Images show a J8-GECI cell interacting with an SLB2 presenting pMHC[null] plus pMHC[9V-hi] (i.e., ~100 molecules of pMHC[9V]/μm²). The different stages were identified by simultaneously imaging close contacts (black holes in the SLB gly-cocalyx fluorescence), cell footprint (cell membrane area), and triggering state (calcium signal) of a cell. A cartoon of each stage is indicated below. See main text for stage descriptions. **b**–**e** Image-based analysis of J8-GECI cells on an SLB2 pre-senting pMHC[null] ± pMHC[9V-lo] or pMHC[9V-hi]. The baseline response is given by non-signaling cells on pMHC[null]. See Supplementary Movies 1, 4, and 5. **b** Cumulative distribution of the searching to scanning stage transition for J8-GECI cells on SLB2s. The analysis uses both signaling and non-signaling cells; $n = 26$ (pMHC[9V-hi]), 26 (pMHC[9V-lo]), and 34 (pMHC[null]) cells. Plotted is the cumulative distribution function of the Kaplan–Meier estimator with the exponential Greenwood confidence inter-val. A pairwise log-rank test indicated that there were no significant differences. **c** Cell footprint versus time, plotted relative to the first appearance of a close contact (timepoint 0 s). Plotted is the mean (±S.D.); $n = 18$ (pMHC[9V-hi], 5 SLBs), 17 (pMHC[9V-lo], 5 SLBs), and 11 (pMHC[null], 6 SLBs) signaling cells, and 23 (pMHC[null], 6 SLBs) non-signaling cells. **d** Number of close contacts versus time. Plotted is the mean (±S.D.). The analysis uses the same cells as in (**c**). **e** Area and diameter of individual close contacts within the first 10 s after their formation. Plotted is the mean (±S.D.). The analysis uses the same cells as in (**c**). **f**–**i** Same analysis as in (**b**–**e**) for primary CD8[+] T cells. UCHT-1 Fab-HaloTag was used as an agonist. See Sup-plementary Movies 6 and 7. **f** Cumulative distribution of the searching to scanning stage transition for primary cells on SLB2s. The analysis uses data for both signaling and non-signaling cells [$n = 19$ (UCHT-1 Fab-HaloTag) and 28 (pMHC[null]) cells]. **g** Cell footprint versus time, plotted relative to the first appearance of a close contact (timepoint 0 s). Plotted is the mean (±S.D.); $n = 8$ signaling cells (UCHT-1 Fab-HaloTag, 3 SLBs) and 14 non-signaling cells (pMHC[null], 6 SLBs). **h** Number of close contacts versus time. Plotted is the mean (±S.D.). The analysis uses the same cells as in (**g**). **i** Area and diameter of individual close contacts within the first 10 s after their formation. Plotted is the mean (±S.D.). The analysis uses the same cells as in (**g**). **j, k** J8-GECI cells interacting with SLB2s presenting pMHC[null] plus pMHC[9V-hi]. **j** Detection of pMHC from a single close contact at calcium release. See Supple-mentary Movie 8. Images are representative of J8-GECI cells for $n = 3$ independent SLBs. **k** Accumulation of ZAP70 at close contacts prior to calcium release. See Supplementary Movie 9. Images are representative of J8-GECI cells for $n = 3$ inde-pendent SLBs. **l** Total area of close contacts versus total cell membrane area at calcium release. 'UCHT-1' refers to UCHT-1 Fab-HaloTag. **m** Number of close con-tacts at calcium release. In **l**, **m**, boxplots indicate the quartiles with a line at the median. Whiskers extend to points that lie within 1.5 IQRs of the lower and upper quartile. Conditions were compared using the Kruskal–Wallis H test, and, if $p < 0.05$, further compared using the pairwise Mann–Whitney U test. The $p$ values are shown. The analysis uses the same cells as in (**c**) for J8-GECI cells and (**g**) for primary cells. **n** Detection of single bound pMHC[9V] at close contacts prior to ($t = -4$ s) and at calcium release ($t = 0$ s). Images show a J8-GECI cell interacting with an SLB2 presenting pMHC[null] plus pMHC[9V-lo]. The arrow indicates a single pMHC[9V] molecule at a close contact. **o** Maximum number of pMHC[9V] bound per close contact per cell for signaling and non-signaling cells; $n = 13$ FOVs, with 8 calcium signaling cells. Source data are provided in the Source data file.

each contact were correlated ($r = 0.75$), suggesting that close-contact persistence depended on the level of CD2/CD58 complex accumulation (Supplementary Fig. 3h). These observations indicate that the TCR is dispensable for close-contact formation and stabilization.

To understand how close contacts are stabilized, we imaged CD58 and ICAM-1 accumulation on pMHC[9V-lo]-presenting SLB2s, to mimic rare agonist presentation in vivo. For J8-GECI cells, we observed that CD58 accumulated exclusively within, and stably tracked with, close contacts during the scanning to synapsing stages (Fig. 4a; Supple-mentary Movie 12). The TCR colocalized in regions of CD58 accumu-lation (Supplementary Movie 13), suggesting that CD2 would be well placed to enhance pMHC detection at close contacts, by controlling membrane separation at the contact (since CD2/CD58 and TCR/pMHC complexes have similar dimensions[51,52]). In contrast, ICAM-1 was stably excluded from the close contacts (Fig. 4b; Supplementary Movie 14)[2] and regions of CD58 accumulation (Supplementary Movie 15), as observed elsewhere[53,54]. The levels of glycocalyx exclusion on the SLB, measured as reductions in fluorescence, were greater in regions of CD58 versus ICAM-1 accumulation (45% versus 20%), indicating that CD2 and CD58 engagement formed tighter contacts (Supplementary Fig. 4a), correlating with the different dimensions of each complex. During the scanning stage, the sequence of events involving the adhesive proteins was: (1) close-contact formation/CD58 accumula-tion, (2) ICAM-1 exclusion, (3) ICAM-1 accumulation in rings around regions of CD58 accumulation, i.e., 'micro adhesion rings', and (4) calcium release (Supplementary Fig. 4b). Similar temporal patterns of CD58 and ICAM-1 accumulation occurred for primary CD8[+] T cells (Supplementary Movie 16).

To distinguish between the effects of CD58 and ICAM-1 on the searching and scanning stages, we removed each protein from pMHC[null]-presenting SLB2s. For J8-GECI cells and in the absence of CD58 on the bilayer, the glycocalyx was penetrated by microvilli (Fig. 4c), but the contacts were less stable (Supplementary Movie 5 versus Supplementary Movie 17), less tight (measured as glycocalyx exclusion: 32% versus 41%; Fig. 4d), and smaller (mean diameter of 0.33 versus 0.44 μm; Fig. 4e). Primary CD8[+] T cells exhibited similar behavior (Fig. 4f–h; Supplementary Movie 7 versus Supplementary Movie 18). We observed similar trends with CD58 removal for events beyond the scanning stage for J8-GECI cells activated on pMHC[9V-lo]-presenting SLB2s (Supplementary Fig. 4c), and for primary T cells (Supplementary Fig. 4d). In contrast, the removal of ICAM-1 from pMHC[null]-presenting SLB2s had little impact on the scanning stage of close-contact formation (see Supplementary Movies 19 and 20 for J8-GECI and primary cells, respectively), but profoundly impacted the spreading and synapsing behavior of cells that pro-duced signaling on pMHC[9V-lo]- and pMHC[null]-presenting SLB2s: fol-lowing calcium release, the cell footprint was greatly reduced (Supplementary Fig. 4c, d).

To relate differences in contact formation and stability during the scanning stage to functional outcomes, we monitored J8-GECI cell calcium responses to pMHC[9V-lo]-presenting SLB2s from which CD58 or ICAM-1 were removed. Consistent with a need for more stable and perhaps larger contacts to better detect pMHC, removal of CD58 caused a substantial reduction in the fraction of responding cells (34% versus 76%; Fig. 4i). In the absence of CD58 the cells exhibited greater displacement, i.e., the cells 'drifted' across the SLB, consistent with CD2 enhancing SLB adhesion generally (Fig. 4j, k). A much smaller reduction in signaling occurred upon ICAM-1 removal (60% versus 76%), likely reflecting a small overall decrease in adhesion to the SLB, since the properties of the close contacts were unchanged. As in the case of pMHC[null]-presenting SLB2s (Fig. 1f), the absence of both adhesion molecules completely blocked signaling and adhesion to the SLB. Notably, the impact on pMHC detection and adhesion, of the loss of CD58 or ICAM-1, was only apparent in the presence of the model glycocalyx, highlighting the important interplay between adhesion proteins and glycocalyx barriers. Collectively, these data indicate that CD2 enhances close-contact formation and stabilization, facilitating antigen detection, whereas ICAM-1/LFA-1 interactions influence the cellular footprint, i.e., degree of spreading, following calcium signaling.

## CD2 enhances CD45 exclusion at close contacts, increasing antigen sensitivity

We recently proposed that sensitive, specific TCR signaling depends on the time the receptor spends in contacts that exclude cellular CD45 (the dwell-time model of T-cell signaling)[3,54]. It seemed likely that CD2 would make important contributions to signaling in this context as CD45 on J8-GECI cells was excluded from areas of CD58 accumulation (Supplementary Fig. 5a)[55]. To explore the function of CD2 in TCR sig-naling, we created a CD2-deficient J8-GECI cell line (J8-GECI-CD2KO)

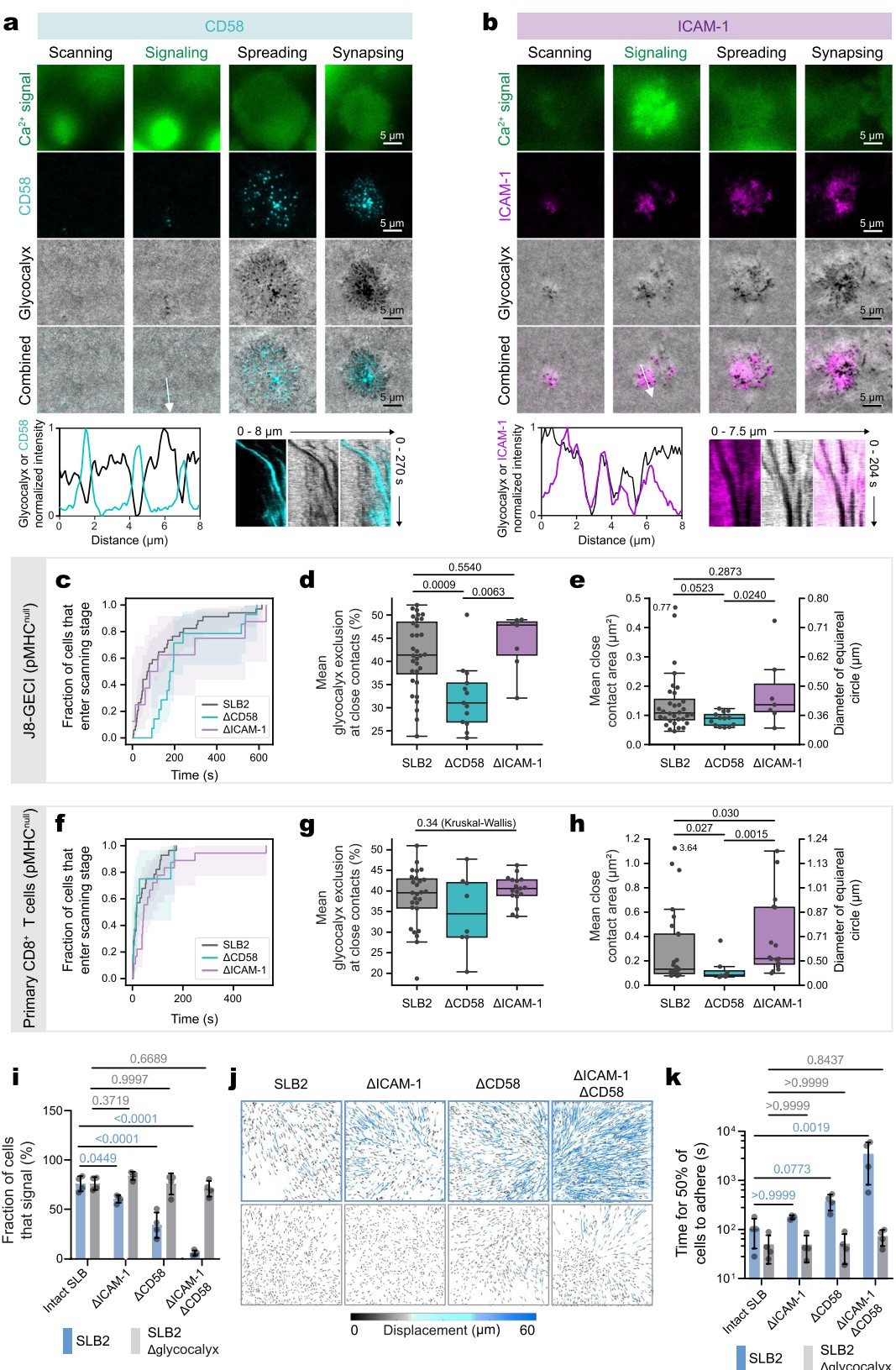

and re-introduced at endogenous levels either wild-type CD2 (J8-GECI-CD2WT), or CD2 whose cytosolic domain was deleted (J8-GECI-CD2ΔCYT). J8-GECI-CD2WT, J8-GECI-CD2ΔCYT, and J8-GECI-CD2KO cells were incubated with fluorescently labeled anti-CD45 antibody and tested for close-contact formation, CD45 exclusion, and calcium release (Supplementary Movies 21–23). To focus on the events leading

to T-cell signaling, we restricted our analysis to the scanning stage of close-contact formation by using pMHC^null-presenting SLB2s.

J8-GECI-CD2WT-expressing cells formed stable close contacts that locally excluded CD45 prior to, and in the absence of, calcium release (Fig. 5a). The numbers of close contacts formed, their mean area, and average glycocalyx exclusion levels were similar for J8-GECI-CD2WT-

**Fig. 4 | CD2 stabilizes close contacts, enhancing antigen detection. a, b** Image-based analysis of J8-GECI cells on an SLB2 presenting pMHC$^{null}$ ± pMHC$^{9V-lo}$. **a** CD58 accumulation occurs at close contacts formed by J8-GECI cells. The min/max normalized intensity line profile was taken along the direction of the white arrow and indicates CD58 accumulation at close contacts. The kymograph indicates persistence of CD58 accumulation spots/close contacts. See Supplementary Movie 12. Images are representative of J8-GECI cells on $n = 3$ independent SLBs. **b** Engaged ICAM-1 is excluded from close contacts formed by J8-GECI cells. The min/max normalized intensity line profile was taken along the direction of the white arrow and indicates ICAM-1 exclusion from close contacts. The kymograph indicates persistent exclusion of ICAM-1 from close contacts. See Supplementary Movie 14. Images are representative of J8-GECI cells on $n = 3$ independent SLBs. **c–e** Close-contact analysis for J8-GECI cells interacting with an SLB2, SLB2ΔCD58, or SLB2ΔICAM-1 presenting pMHC$^{null}$; $n = 34$ (SLB2, the same SLBs as in Fig. 3b–e), 14 (ΔCD58, from 4 SLBs), and 8 (ΔICAM-1, from 4 SLBs) cells. See Supplementary Movies 5, 17, and 19. **c** Cumulative distribution of the searching to scanning stage transition for cells on the indicated SLBs. The analysis uses data for both signaling and non-signaling cells. Plotted is the cumulative distribution function of the Kaplan–Meier estimator with the exponential Greenwood confidence interval. Data were examined using a pairwise log-rank test, which indicated that there were no significant differences. **d** Mean glycocalyx exclusion (i.e. contact tightness) at close

contacts during the scanning stage for each cell. **e** Mean close-contact area during the scanning stage for each cell. The analysis uses the same cells as in (**d**). In **d** and **e**, conditions were compared using the pairwise Mann–Whitney U test. The $p$ values are shown. The boxplots indicate the quartiles with a line at the median. Whiskers extend to points that lie within 1.5 IQRs of the lower and upper quartile. **f–h** Same analysis as in (**c–e**) using human primary CD8$^+$ T cells; $n = 28$ (SLB2, the same SLBs as in Fig. 3f–i), 8 (ΔCD58, from 3 SLBs), and 18 (ΔICAM-1, from 3 SLBs) cells. See Supplementary Movies 7, 18, and 20. In **f**, a pairwise log-rank test indicated that there were no significant differences. **i** Fraction of J8-GECI cells that signal. Shown is the mean ± S.D. of $n = 4$ SLBs per condition with ≥257 cells analyzed per SLB. SLB2 data (blue bars) and SLB2Δglycocalyx (gray bars) were compared with a one-way ANOVA with Dunnett correction using data for the 'intact' SLBs, i.e., SLB2s with and without glycocalyces, as the control groups. **j** Displacement tracks. Each dot/line represents a single J8-GECI cell tracked for up to 10 min, exploiting the currents generated by adding cells to the SLBs to probe the adhesiveness of the different SLBs. Data are from the same experiment as in (**i**). **k** The time taken for 50% of J8-GECI cells to adhere to the SLBs. SLB2 data (blue bars) and SLB2Δglycocalyx (gray bars) were compared separately with a one-way Kruskal–Wallis test with Dunn's correction using data for the intact SLBs as the control groups. Data are from the same experiment as in (**i**). Source data are provided in the Source data file.

and J8-GECI-CD2ΔCYT-expressing cells (Fig. 5b–d). J8-GECI-CD2WT- and J8-GECI-CD2ΔCYT-expressing cells excluded ~40% and ~32% of CD45 from close contacts, respectively (Fig. 5e). In marked contrast, J8-GECI-CD2KO cells formed smaller close contacts (Fig. 5c) that were characterized by significantly reduced exclusion of the glycocalyx (Fig. 5d), consistent with the effects of removing CD58 from SLB2s (Fig. 4c–h). Surprisingly, for J8-GECI-CD2KO cells, CD45 exclusion at each contact was similar to that for J8-GECI-CD2ΔCYT-expressing cells (~33%; Fig. 5e), despite the contacts being smaller. EC$_{50}$ values for calcium signaling by J8-GECI-CD2WT-, J8-GECI-CD2ΔCYT-expressing cells, and for J8-GECI-CD2KO cells on SLB2s presenting pMHC$^{null}$ and increasing amounts of pMHC$^{9v}$, were 0.7, 3.2, and 64.5 pMHC$^{9v}$ molecules/μm$^2$, respectively (Fig. 5f). The differences were CD2 dependent, as the cell lines expressed other key surface proteins comparably (Supplementary Fig. 5b) and responded equally well to direct TCR stimulation with an activating antibody (OKT3; Supplementary Fig. 5c). The slight increase in the sensitivity of J8-GECI-CD2WT- versus J8-GECI-CD2ΔCYT-expressing cells likely reflects the slightly lower CD45 exclusion by J8-GECI-CD2ΔCYT-expressing cells and/or the enhanced recruitment of Lck to the contact by full-length CD2 (Fig. 5g; Supplementary Fig. 5d). These results indicate that CD2 has a pivotal role in stabilizing close contacts and altering the kinase/phosphatase balance at these sites.

## Dependence of TCR discrimination on constrained close-contact size

Finally, we examined the relationship between close contact formation and TCR ligand discrimination. J8-GECI cells were capable of efficient ligand discrimination, with EC$_{50}$ values for responses to pMHC with affinities ranging from 7 μM to >2000 μM varying by three orders of magnitude (Fig. 6a), all within a narrow time window of 100–200 s (Fig. 6b). Using low densities of agonist pMHC (1 molecule/μm$^2$), the transition to the scanning stage was found to be independent of pMHC affinity (Supplementary Fig. 6a), whereas transitioning from scanning to spreading was affinity dependent, reflecting discrimination (Supplementary Fig. 6a). Once again, irrespective of ligand affinity, or the delay before signaling, close-contact diameter was constrained to the small range of 0.32–0.43 μm at calcium release (Fig. 6c), with no difference in the degree of glycocalyx exclusion at the contacts (Supplementary Fig. 6b). Simulations confirmed that at the resolution limit of the microscope smaller contacts would have been reliably detected, emphasizing the uniformity and constrained size of the contacts (Supplementary Fig. 6c).

To demonstrate the need for constrained contacts, as predicted by the dwell-time model of signaling[3], we examined the effects of increasing contact size on pMHC discrimination. J8-GECI-CD2KO cells overexpressing CD2 by a factor of ~10 (J8-GECI-CD2WT$^{hi}$; Supplementary Fig. 6d) produced ~5–10-fold larger close contacts, on average, than J8-GECI-CD2WT-expressing cells (Fig. 6d, Supplementary Fig. 6e). Notably, J8-GECI-CD2KO, J8-GECI-CD2WT, and J8-GECI-CD2WT$^{hi}$ cells on pMHC$^{null}$-presenting SLB2s produced calcium responses of 5%, 20%, and 65%, respectively (Fig. 6e), revealing a loss of discrimination as contact size increases. Receptor expression levels (Supplementary Fig. 6d), and levels of activation via direct engagement of the TCR (Supplementary Fig. 6f) were comparable for all cell lines, suggesting that the signaling differences were dependent on the extent of CD2 engagement. A TCR-deficient J8-GECI-CD2WT$^{hi}$ cell line did not produce calcium signals on an SLB2 presenting pMHC$^{null}$ (Fig. 6e) or via direct TCR engagement (Supplementary Fig. 6f), indicating that the signaling observed at high levels of CD2 expression was TCR-dependent.

To further confirm that ligand discrimination was lost by J8-GECI-CD2WT$^{hi}$ cells, we examined the effect of varying pMHC affinity (7–3200 μM K$_D$) at constant pMHC density, on the signaling responses. The larger contacts formed by J8-GECI-CD2WT$^{hi}$ cells could be readily segmented using our automated analysis pipeline (Supplementary Fig. 6g). Using SLB2s presenting pMHC$^{null}$ and ~10 molecules/μm$^2$ of pMHC varying in affinity, we observed that contact size was unaffected by pMHC affinity or the presence of the TCR (Supplementary Fig. 6h), and that ~70% of J8-GECI-CD2WT$^{hi}$ cells produced calcium responses irrespective of pMHC affinity (Fig. 6f, Supplementary Movie 24). This was in marked contrast to cells expressing native levels of CD2, which formed much smaller contacts and efficiently discriminated between different pMHC at this density (Fig. 6f). These results indicate that contact size underpins T-cell ligand discrimination.

## Discussion

Much of what we know about T-cell interactions with apposing surfaces has come from studies of T cells interacting with SLBs presenting both TCR and integrin (i.e., LFA-1) ligands. As invaluable as this approach has been, the interplay between small adhesion proteins and the target cell glycocalyx, and their contributions to antigen recognition and TCR signaling, could not be studied. More importantly, alongside this and until recently, the role of microvilli in contact formation and T-cell responsiveness has received limited attention.

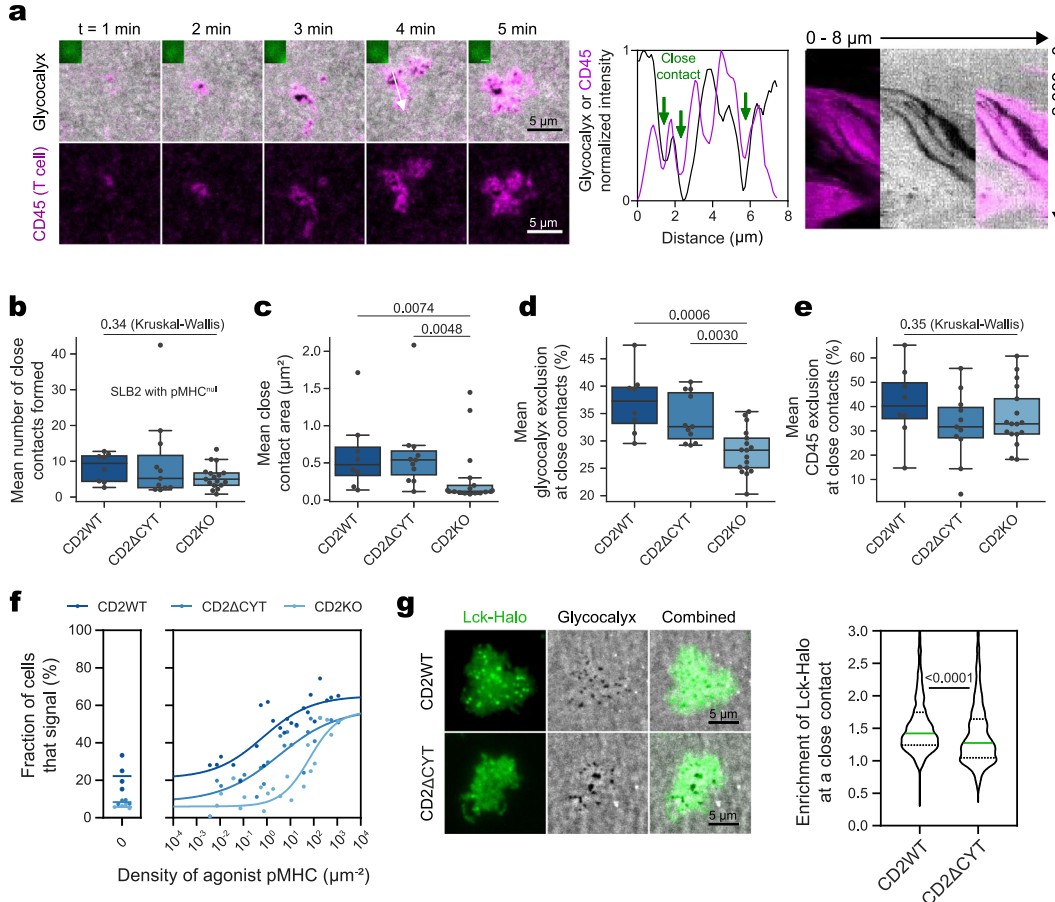

**Fig. 5 | CD2 enhances T-cell sensitivity. a** CD45 exclusion at close contacts formed by J8-GECI-CD2WT cells. The images show a J8-GECI-CD2WT cell during the scanning stage on an SLB2 presenting pMHC^null; inset shows GECI fluorescence, indicating a lack of signaling. A min/max normalized intensity line profile was taken along the direction of the white arrow and indicates CD45 exclusion at close contacts. The kymograph shows contact stabilization and persistent CD45 exclusion from contacts. See Supplementary Movies 21–23. **b**−**e** Close contact feature analysis for both signaling and non-signaling J8-GECI-CD2WT and variant cells during the scanning stage on an SLB2 presenting pMHC^null; $n = 9$ (J8-GECI-CD2WT, from 2 SLBs), 11 (J8-GECI-CD2ΔCYT, from 2 SLBs), and 17 (J8-GECI-CD2KO, from 3 SLBs) cells. The boxplots indicate the quartiles with a line at the median. Whiskers extend to points that lie within 1.5 IQRs of the lower and upper quartile. Conditions were compared using the Kruskal–Wallis H test, and, if $p < 0.05$, further compared using the pairwise Mann–Whitney U test. The $p$ values are shown. **b** Mean number of close contacts per cell. **c** Mean single close-contact area. The analysis uses the same cells

as in (**b**). **d** Mean glycocalyx exclusion at close contacts. The analysis uses the same cells as in (**b**). **e** Mean exclusion from close contacts of CD45 on the J8-GECI cell surface. The analysis uses the same cells as in (**b**). **f** Calcium response curves for J8-GECI-CD2WT, J8-GECI-CD2ΔCYT, and J8-GECI-CD2KO cell lines interacting with SLB2s presenting pMHC^null ± pMHC^9V at the indicated densities. Data were fitted using a four-point dose-response curve, constrained to a minimum based on responses to pMHC^null; $n = 27$ (J8-GECI-CD2WT), 25 (J8-GECI-CD2ΔCYT), and 25 (J8-GECI-CD2KO) SLBs per curve with ≥139 cells analyzed per SLB. The $EC_{50}$ values for J8-GECI-CD2WT, J8-GECI-CD2ΔCYT, and J8-GECI-CD2KO cells were 0.7, 3.2, and 64.5 molecules pMHC^9V/μm², respectively. **g** Increased Lck recruitment to close contacts for cells expressing full-length CD2 versus CD2ΔCYT on an SLB2 presenting pMHC^null + pMHC^9V-lo; $n = 1080$ (J8-GECI-CD2WT) and 983 (J8-GECI-CD2ΔCYT) contacts analyzed from 10 FOVs. Means were compared using a two-sided Student's t-test. Source data are provided in the Source data file.

We implemented a SLB-based surrogate of the APC surface, comprising the major glycoprotein elements of the glycocalyx formed by APCs, the ligands of small and large adhesion proteins, and both model self and agonistic pMHC. We discovered that, in this more physiological setting, T-cell responses were profoundly affected by the presence of a glycocalyx and the small adhesion molecules. Removing the model glycocalyx provoked strong signaling and a loss of T-cell specificity. Conversely, eliminating the adhesive proteins reduced and delayed signaling, confirming the barrier-like activity of the cell glycocalyx[5]. T cells overcame the model glycocalyx by 'punching through' the barrier, forming numerous contacts visualized as small black holes in the layer of fluorescently labeled CD45 and CD43 molecules on the SLB. We referred to these structures as 'close contacts'. Contact formation was an active process that proceeded in four stages, i.e., sequential 'searching' and 'scanning' modes separated by the formation of close contacts, a TCR- and pMHC-dependent transition to a 'spreading' stage, and, finally, synapse formation. Each

close contact was stabilized by the interaction of CD2 with CD58; removing CD58 reduced the sensitivity of recognition ~100-fold. Sensitivity was enhanced by CD2 via increased close-contact persistence, CD45 exclusion over a larger area, and increased Lck recruitment within close contacts. ICAM-1, on the other hand, which was excluded from close contacts, did not enhance antigen sensitivity or early close-contact formation but was required during the spreading stage following antigen recognition, presumably when LFA-1 'switches' to a high-affinity state (reviewed in ref. [56]). These results indicate that the primary adhesive protein functioning during antigen recognition is CD2 and explain how naïve T cells can respond to DCs in an antigen-dependent/ICAM independent manner[57]. CD2 is reported to be enriched at microvillar tips, which would potentiate CD2 engagement during initial contacts[58]. Increasing the glycocalyx density three-fold compromised the efficiency of the T-cell response, consistent with the effects, e.g., of the altered composition of the cancer cell glycocalyx[5]. We note that such effects could be further

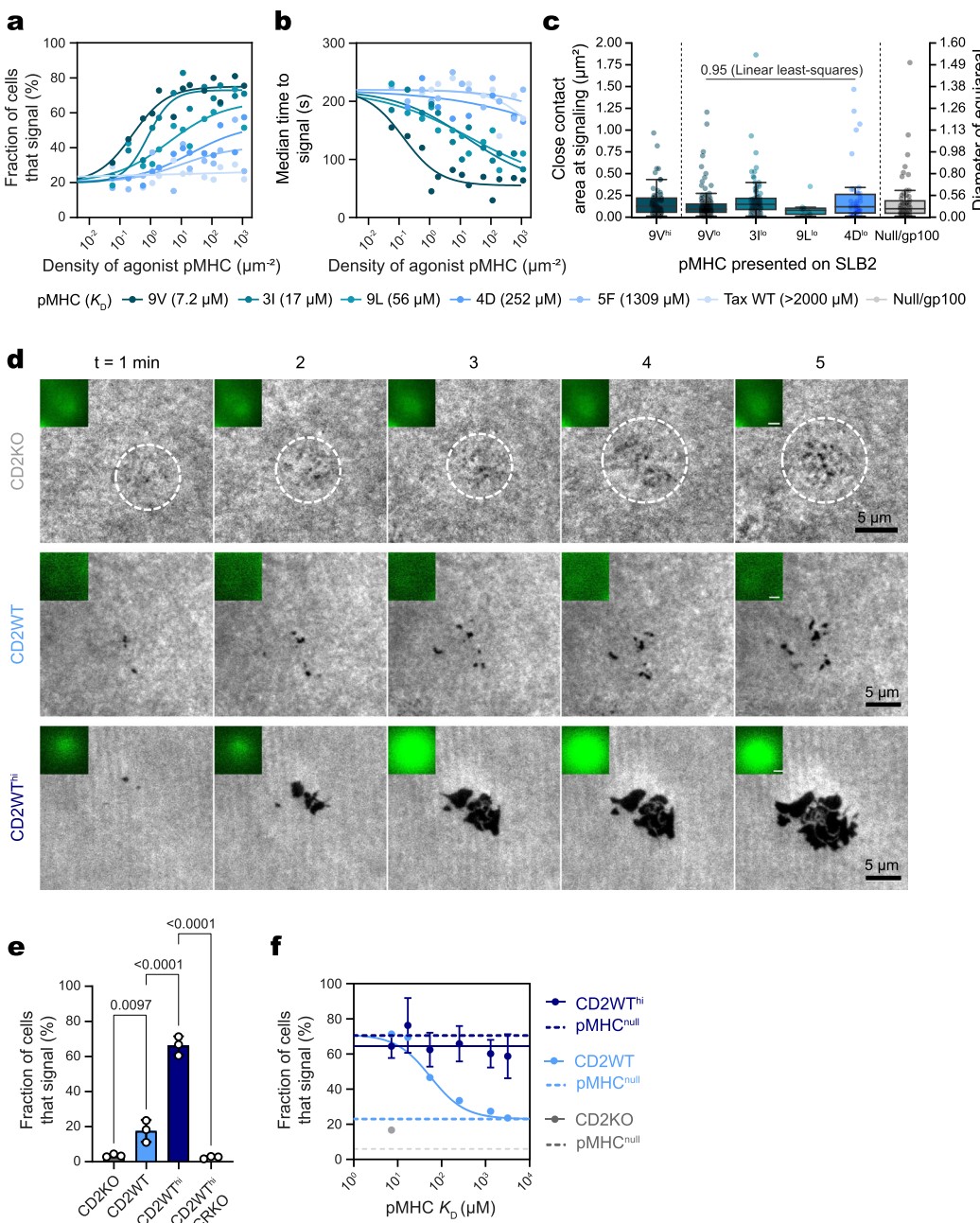

**Fig. 6 | Antigen discrimination requires size-constrained close-contact formation. a** Calcium response curves for J8-GECI cells interacting with SLB2s presenting pMHC[null] plus increasing densities of different-affinity pMHC. **b** Median time taken for cells to signal. From the same experiment as in (**a**). In **a**, **b**, each datapoint corresponds to a separate SLB; $n = 14$ SLBs with ≥148 cells analyzed per SLB. Data were fitted using a four-point dose-response curve, with the bottom of each curve in (**a**), and top of each curve in (**b**), constrained to responses on an SLB2 presenting pMHC[null]. **c** Area (left y-axis) and diameter (right y-axis) of close contacts formed by J8-GECI cells interacting with an SLB2 presenting pMHC[null] plus the indicated agonist pMHC at point of calcium release; $n = 68$ (9V[hi], from 18 cells), 148 (9V[lo], from 17 cells), 100 (3I[lo], from 16 cells), 9 (9L[lo], from 5 cells), 42 (4D[lo], from 7 cells), and 58 (Null/gp100, from 11 cells) individual contacts. Data obtained for pMHC[null] and pMHC[9V-hi/lo] SLBs are from the same experiment as in Fig. 3b–e. To measure the correlation between the affinity of low-density pMHC and the contact size, a linear least-squares regression was performed, and the p-value shown. The

boxplots indicate the quartiles with a line at the median. Whiskers extend to points that lie within 1.5 IQRs of the lower and upper quartile. **d** TIRFM images of close-contact formation with/without calcium release for the indicated cell lines on SLB2s presenting pMHC[null]. A white dashed circle is shown for the J8-GECI-CD2KO cell line to indicate the region of close-contact formation. **e** Fraction of cells that produce signals on an SLB2 presenting pMHC[null]. Shown is the mean (± S.D.) of $n = 3$ SLBs with ≥309 cells analyzed per SLB. Means were compared using one-way ANOVA with Šídák correction. **f** Fraction of cells that signal on an SLB2 presenting pMHC[null] plus ~10 molecules/μm² of different-affinity pMHC. Data obtained for J8-GECI-CD2WT[hi] cells are shown in dark blue with each point representing the mean ± S.D ($n = 3$ SLBs per agonist pMHC with 16-75 cells analyzed per SLB). The light blue line is taken from curves fitted in (**a**) for data obtained with the J8-GECI cells. The gray datapoint is taken from Fig. 5f for the J8-GECI-CD2KO cell line. Source data are provided in the Source data file.

exaggerated for deeper as well as denser glycocalyces. A remarkable feature of close contacts formed under all conditions, including for bilayers with more dense glycocalyces, however, was their uniformity, each being confined to ~0.1 μm² (~0.36 μm diameter) prior to signaling. Most importantly, a dramatic loss of TCR discrimination was observed when close-contact size was larger than ~0.1 μm².

Previously, Cai et al. used SLBs presenting fluorescent quantum dots to study T-cell contact formation, a method they called synaptic contact mapping (SCM)[2]. Using SCM, they observed that 0.5 μm holes formed in the quantum dot layer. The surfaces of the T cells we studied were populated with numerous microvilli, and we assume that the contacts observed using SCM are analogous to the close contacts formed on our bilayers. As in our experiments, LFA-1 and ICAM-1 were excluded from the close contacts, but Cai et al. also observed that contact stabilization was TCR-dependent. In contrast, we observed that contact stabilization was TCR-independent, underscoring the important contribution of small adhesion proteins to antigen detection. Since the ligands of small adhesive proteins were not present in the bilayers used for SCM, TCR/pMHC interactions likely performed the important function of physically stabilizing contact formation in the experiments of Cai et al. Given that, in our experiments, TCR-deficient T cells formed close contacts normally but did not progress to spreading, the scanning-to-spreading transition must comprise a key early checkpoint, whereupon the cells will have already discriminated between different-affinity pMHC[19]. Remarkably, we found that a single close contact was sufficient for this transition. During the spreading stage, T cells were observed to form many additional contacts, i.e., three-fold more, with the SLB. This 'intensified interrogation' of the activating surface likely allows T cells to reach higher activation thresholds (e.g., NFAT translocation). Finally, whereas Cai et al. did not observe "profound" exclusion of CD45, 30–40% local exclusion of CD45 at close contacts was readily demonstrable in our experiments. We note, however, that it is also proposed that CD45 is "pre-excluded" from the tips of microvilli[39].

Why do T cells use structures as exotic as microvilli to engage target cells? It could have been expected that the formation of intimate contacts over large areas would allow for more efficient and rapid antigen detection. One important factor is that the glycocalyx barrier needs to be overcome, which might be more easily achieved with small protrusions. An additional requirement, we propose, is the need for T cells to be discriminatory. We have shown that at physiological levels of CD2 expression and in the presence of a glycocalyx, T cells could discriminate readily between ligands varying in affinity from 7 μM to >2000 μM $K_D$, using close contacts of ~0.1 μm² (~0.36 μm diameter). However, increasing contact size by overexpressing CD2 led to a dramatic loss of antigen discrimination. But how can the TCR be triggered in the absence of strongly-binding ligands? Computational simulations based on the dwell-time model of TCR triggering have suggested that if large close contacts form, TCRs are unlikely to encounter phosphatases, favoring receptor phosphorylation and signaling[3]. In contrast, when contacts are small, i.e., ~0.1 μm², the simulations showed that discriminatory signaling could be readily achieved. Indeed, the simulations predicted the relative potencies of pMHC ligands presented to CD4+ and CD8+ T cells with remarkable accuracy. Strongly supporting this treatment of receptor triggering, slowing the diffusion of the TCR suffices to initiate signaling[54]. Here, we directly confirmed that ligand discrimination by T cells is strictly reliant on the formation of CD2-stabilized, microvillus-sized contacts. Given these findings, it is conceivable that the immune system varies CD2 expression to tune responsiveness. Memory T cells have enhanced sensitivity to lower-affinity antigens which is perhaps explained by their higher expression of CD2[60,61]. However, the large CD2-dependent contact that forms at mature synapses called the "corolla", which is thought to amplify responses[37], is unlikely also to contribute to discriminatory TCR signaling.

This study has explored the relationship between small and large adhesion molecules, a cell glycocalyx, close-contact formation, and self/non-self discrimination by T cells interacting with glass-supported bilayers facilitating imaging. Our findings need now to be confirmed for bona fide T cell/APC contact or for interactions of T cells with even better models of APC surfaces. We previously argued that the appearance of a phosphatase-containing glycocalyx on primitive lymphoid cells during their evolution may have allowed their utilization of a contact-sensitive, molecular segregation-based mechanism of receptor signaling[49]. Our data suggest that the use of microvilli could have been a necessary, complementary adaptation which ensured that individual contacts were small in area, allowing discriminatory signaling by these primitive cells.

## Methods

### Cell culture
All cell culture was performed in HEPA-filtered cell culture cabinets. All media components were bought sterile or 0.22 μm filtered before use. Cells were grown at 37 °C in 5% CO₂. Cell density and viability were monitored with a 1:1 mix of cell culture medium and 0.4% Trypan blue. Stained cells were analyzed with a Countess II automated cell counter (ThermoFisher). Cells were independently tested negative for mycoplasma (Human Immunology Unit, WIMM). Jurkat T cells and derived cell lines were cultured in RPMI-1640 medium supplemented with 10% (v/v) fetal calf serum (FCS), 1% (v/v) HEPES buffer, and 1% (v/v) pen/strep/neo antibiotics (Sigma; complete RPMI). Cells were maintained between 0.1–1 × 10⁶ ml⁻¹. Human embryonic kidney 293T (HEK-293T) cells were cultured in Dulbecco's modified Eagle medium (DMEM) supplemented with 10% (v/v) FCS, 1% (v/v) pen/strep/neo antibiotics (Sigma), and 1% (v/v) glutamine (complete DMEM). The catalog numbers for commercial items can be found in Supplementary Table 1.

### Creation of cell lines
The J8 and TCRKO cell lines were created from Jurkats (clone E6-1) as previously described[42,50]. ZAP70-HaloTag was expressed in J8 cells via lentiviral transduction to produce the J8-ZAP70-Halo cell line. Expression of the jGCaMP7s[43] in the J8 (referred to as J8-GECI) or TCRKO (referred to as TCRKO-GECI) cell lines was undertaken and validated as previously described[50,54]. To ablate CD2 in J8-GECI and TCRKO-GECI cells, CRISPR-Cas9 guides were designed and selected for high specificity with minimal off-targets using Benchling (hg38 reference genome). Oligonucleotides were then designed, annealed, and ligated into the LentiCRISPRv2 plasmid using the dual BsmBI sites, as previously described[62,63]. Cells were sorted on the CD2-negative population by FACS 7 days after lentiviral transduction to produce either J8-GECI-CD2KO or TCRKO-GECI-CD2KO cells. To produce the J8-GECI-CD2WT, J8-GECI-CD2WThi, and TCRKO-GECI-CD2WThi cell lines, J8-GECI-CD2KO or TCRKO-GECI-CD2KO cells were stably transduced with wild-type CD2 (CD2WT) cDNA or CD2 cDNA encoding a cytosolic domain deletion leaving only 3 aa of the intracellular domain (CD2ΔCYT), using lentiviruses. Cells were sorted for endogenous or high levels of CD2WT or CD2ΔCYT expression by FACS. Lck-HaloTag was inserted into CD2WT and CD2ΔCYT by lentiviral transduction to produce the J8-GECI-CD2WT-Lck-Halo and J8-GECI-CD2ΔCYT-Lck-Halo cell lines. For all cells used in this study, LFA-1 expression was increased by lentiviral expression of cDNA encoding CD11a and CD18 to match the expression of these proteins in human primary CD8+ T cells that were primed and rested.

### Lentivirus production and transduction
1 × 10⁶ HEKs in complete DMEM were seeded onto 6-well plates 24 h prior to the addition of plasmids. After 24 h, 0.5 μg of pHR/Lenti-CRISPRv2-based plasmid was co-transfected with vectors containing the lentiviral packaging proteins (0.5 μg of p8.91 and 0.5 μg pMDG). GeneJuice (Merck) was used to transfect the plasmid mixture

according to the manufacturer's protocol for adherent cells. Forty-eight hours post-transfection, supernatant was collected, filtered (0.22 μm), and added to a 6-well plate containing $0.5-1 \times 10^6$ Jurkat cells or derivative cell lines in 2 ml complete RPMI. Forty-eight hours post infection 4 ml of complete RPMI was added. Cells were analyzed at least 72 h post transduction. After 72 h, CRISPR KO cells were further selected by treatment with 1 ng/μl puromycin in complete RPMI for 3 days and/or cells were sorted by FACS after 7 days.

## Human primary CD8$^+$ T cells

T cells were obtained from blood leukocyte cones purchased from NHS Blood and Transplant, John Radcliffe Hospital, Oxford, UK. Blood cones were used under the ethical guidelines of NHS Blood and Transplant. CD8$^+$ T cells were isolated by Ficoll-Paque density gradient centrifugation and the use of a CD8$^+$ T Cell Isolation Kit (Miltenyi) according to the manufacturer's protocol. Isolated CD8$^+$ T cells were washed and resuspended in complete RPMI with IL-2 (50 U/ml, PeproTech) and CD3/CD28-coated Human T-Activator Dynabeads (ThermoFisher Scientific). Cells were resuspended in complete RPMI containing fresh IL-2 every 2 days. Dynabeads were removed after day 5, and cells left to further expand for 7 more days. On day 12, aliquots of cells were frozen for future use. Twenty-four hours before use, cells were thawed, washed, and resuspended in complete RPMI with IL-2 (50 U/ml). On the day of use, dead cells were removed using the Dead Cell Removal Kit (Miltenyi Biotec) according to the manufacturer's protocol and the live cells placed back into complete RPMI and IL-2 (50 U/ml).

## Flow cytometry

Cells were stained using conjugated mouse anti-human antibodies specific for the indicated proteins. The following antibodies were used: Anti-CD2-PE (BioLegend, Cat# 300208, 1:100), Anti-CD3-PE (BioLegend, Cat# 300408, 1:100), Anti-CD4-PE (BioLegend, Cat# 300508, 1:100), Anti-CD8α-PE (BioLegend, Cat# 344706, 1:100), Anti-CD11a-PE (BioLegend, Cat# 350606, 1:100), Anti-CD45-PE (BioLegend, Cat# 304008, 1:100), Anti-HLA-A2-PE (BioLegend, Cat# 343306, 1:100), Anti-B2M-PE (BioLegend, Cat# 316306, 1:100), Anti-CD54-PE (BioLegend, Cat# 353105, 1:100), Anti-CD58-PE (BioLegend, Cat# 330905, 1:100), Anti-CD43-PE (BioLegend, Cat# 343203, 1:100), Anti-CD83-FITC (BioLegend, Cat# 305306, 1:100), Anti-CD1a-Pacific Blue (BioLegend, Cat# 300124, 1:100), Anti-CD14-PerCP/Cyanine 5.5 (BioLegend, Cat# 367110, 1:100), Anti-CD11c-AF647 (BioLegend, Cat# 301620, 1:100), Anti-HLA-DR-APC-Cy7 (BioLegend, Cat# 307618, 1:100), and IgG1 Isotype-PE (MOPC-21; BioLegend, Cat# 400114, dilution matched to the highest other PE-labeled antibody concentration in the staining experiment). $0.5 \times 10^6$ cells were counted, washed in PBS (0.01% NaN$_3$; PBS azide), and then stained at 4 °C for 1 h using the indicated dilution of the antibody. Cells were washed ×2 in PBS azide, fixed in 2% paraformaldehyde (PFA), and analyzed by flow cytometry. Cells were analyzed using the Attune NxT (Lifetechnologies) flow cytometer. Compensation was performed where required. Data were analyzed using FlowJo. All cell sorting was performed by the Weatherall Institute of Molecular Medicine FACS Facility.

## Protein production and labeling

Soluble forms of CD45, CD58, and ICAM-1 (CD54) were made as previously described[64]. Briefly, cDNA encoding the ECD of CD43 (residues 20-253 UniProtKB P16150) was cloned into the pHR plasmid downstream of the sequence encoding cRPTPσSP. The CD43 cDNA was modified to encode a H$_6$-SRAWRHPQFGG-H$_6$ 'H$_6$-spacer-H$_6$' tag at the C-terminus for purification and stable interaction with 1,2-dioleoyl-sn-glycero-3 (DGS)-NTA(Ni$^{2+}$) containing SLBs[65]. Soluble protein expressed by lentiviral transduction of HEK 293T cells was purified using metal-chelate and size-exclusion chromatography with an AKTA Pure protein purification system. pMHC was produced as previously described[66]. Briefly, cDNA encoding the ECD of HLA-A2 (residues 25-304, UniProtKB P79603) and beta-2-microglobulin (β$_2$M, residues 21-119, UniProtKB P61769) were ligated into pET28a (+; kanamycin resistant) vector for expression in Rosetta 2 (DE3)pLysS competent *E. coli* (Merck). The HLA-A2 cDNA was modified with a 'H$_6$-spacer-H$_6$' tag at the C-terminus to allow interaction with SLBs. HLA-A2 and β$_2$M were purified from inclusion bodies and folded in the presence of either 9V, 3I, 9L, 4D, 5F, Tax WT, or gp100 peptide[41,67]. Monomeric pMHC was purified using the AKTA Pure protein purification system. UCHT-1 Fab was prepared from purified antibody using immobilized papain as directed by the manufacturer (ThermoFisher). Fab digestion and purity were confirmed by size exclusion chromatography (AKTA). A C-terminally histidine- and HaloTag-tagged form of the UCHT-1 Fab was expressed by lentiviral transduction in HEK 293T cells and purified using metal-chelate and size-exclusion chromatography (referred to as UCHT-1 Fab-HaloTag). OKT3 antibody was provided by the Medical Research Council Human Immunology Unit, Oxford. All proteins were snap-frozen in dry ice and stored at −80 °C prior to use. Proteins were labeled using either an Alexa Fluor (Alexa)-488, -555, or -647 Antibody Labeling Kit (ThermoFisher) according to the manufacturer's instructions, or a similar custom protocol using the NHS dye derivatives. Briefly, 10% (v/v) 1 M sodium bicarbonate solution was added to the protein and incubated with a 10-fold molar excess of the NHS dye (diluted in anhydrous dimethyl sulfoxide, DMSO) for 1 h, followed by purification in a SEC spin column (Bio-Rad). Proteins were labeled at ≥1 dye per molecule. Prior to use, the frozen protein was thawed at 4 °C, centrifuged ($17,000 \times g$, 5 min, 4 °C) to remove any large aggregates formed during the thawing process, re-aliquoted to a fresh tube, and the concentration determined using a NanoDrop Spectrophotometer (Labtech). After thawing, the protein was labeled, divided into 10 μl aliquots, and snap-frozen or used and stored at 4 °C for 2 weeks before being discarded.

## Protein densities on moDC

Monocyte-derived dendritic cells (moDCs) were produced by isolating human monocytes from PBMCs using a Ficoll gradient and a CD14$^+$ CD16$^-$ Magnetic Bead Isolation Kit (Miltenyi). Purified monocytes were grown in an uncoated 24-well plate at $1 \times 10^6$ ml$^{-1}$ in RPMI 1640 supplemented with 10% (v/v) FCS, 1% (v/v) sodium pyruvate, 1% (v/v) HEPES, 1% (v/v) pen/strep antibiotics, 1% (v/v) L-glutamine, 1% NEAA (v/v), and 1% β-mercaptoethanol at 37 °C with 5% CO$_2$. Monocytes grown in medium without cytokines were used as a negative control. To differentiate monocytes into immature moDCs recombinant human IL-4 (10 ng/ml) and GM-CSF (100 ng/ml; BioLegend) were added to the medium for 1 week, with fresh medium and cytokine mix added on day three of seven. On day 5, spent medium was replaced with fresh medium containing the cytokines and 200 ng/ml of LPS (Sigma-Aldrich) to create mature moDCs. These were analyzed by flow cytometry for mature moDC markers (CD14, CD1a, CD11c, CD83, HLA-DR, and Live/Dead stain[68]) in combination with either β$_2$M, HLA-A2, CD43, CD45, CD58, or ICAM-1 PE-conjugated antibodies (BioLegend). BD Biosciences Quantibrite PE calibration beads were used to estimate total numbers of surface proteins per mature moDC, as per the manufacturer's protocol. To convert total protein per moDC into a density, the surface area of moDCs was calculated from z-stack (3D) images taken using a ZEISS LSM 880 with Airyscan every 200 nm from the basal to apical plane in Airyscan mode. Briefly, moDCs were labeled with CellMask Red (ThermoFisher) according to the manufacturer's protocols, fixed in 4% PFA and 0.25% glutaraldehyde at room temperature for 15 min, then placed on PLL-coated glass. Total surface area was calculated by thresholding and then quantifying the perimeter of the cell membrane (outer line) in each frame of a z-stack, multiplying the perimeter by the depth (i.e., 200 nm) to obtain the surface area per frame, and summing the 'per frame surface area' across the entire

z-stack. This gave a median surface area of ~2000 µm², consistent with other measurements[69,70]. Total protein amounts were divided by the median surface area to derive approximate protein densities.

## Glass-supported lipid bilayers

SLBs were prepared using vesicle fusion. A lipid mixture consisting of 98% (mol%) POPC (Avanti Polar Lipids) and 2% (mol%) DGS-NTA-Ni²⁺ (NiNTA; Avanti Polar Lipids) in chloroform were mixed in a cleaned glass vial and dried under a stream of nitrogen. 2% NiNTA was chosen to minimize unwanted interactions of the lipid with biological material (e.g., non-tag histidine within proteins and negatively charged sugars on cells) whilst allowing physiological densities of multiple his-tagged proteins to be attached to the surface[71]. The dried lipid mix was resuspended in 0.22 µm filtered (0.02 µm filtered for TIRFM experiments) PBS at 1 mg/ml, vortexed, and sonicated (30 min on ice with a tip sonicator, or until transparent in an ultra-sonicator bath) to produce small unilamellar vesicles (SUVs). Glass coverslips (25 mm, thickness no. 1.5; VWR) were cleaned for at least 2 h in 3:1 sulfuric acid/hydrogen peroxide at room temperature, rinsed in MQ water, and plasma cleaned for 1 min (oxygen plasma) or 20 min (argon plasma). CultureWell 50-well silicon covers (Grace Bio-Labs) were cut and placed on the washed coverslips (maximum 4/coverslip). SUVs were added to each well at a final concentration of 0.5 mg/ml (10 µl total volume) and left for 1 h at room temperature. Wells were washed at least five times by removing and adding PBS to each well. The amount of liquid in each well was adjusted to be level with the well edge before adding 5 µl of proteins mixed at the desired concentrations. This was done to ensure reproducible incubation concentrations. Protein mixes were incubated with the bilayer for an hour at room temperature and washed ten times in pre-warmed (37 °C) PBS or PBS plus 2 mM MgSO₄ (37 °C) immediately before use. Point fluorescence correlation spectroscopy (pFCS) was used to relate protein concentration to density on the SLB.

## Point fluorescence correlation spectroscopy

The Zeiss 780 equipped with a ×40 water objective was used for pFCS measurements. Data were acquired in photon counting mode, using a 488/594/633 MBS, and a pinhole set to 1 AU. Excitation was performed with a 633 nm He-Ne laser set at 1% power for Alexa-647 labeled proteins. To calibrate the confocal volume size, 100 nM Alexa-647 dye in PBS was used. SLBs were allowed to acclimate to 37 °C before measurements were taken, and the z-axis was tuned to the highest intensity plane before imaging. Three 10 s measurements were typically taken for each SLB.

Data were fitted and processed using PyCorrFit software[72]. Auto-correlation values G(τ) were fitted to the following equation:

$$G(\tau) = G(0) \cdot GD(\tau) \cdot GT(\tau) + \text{Off} \quad (1)$$

The autocorrelation value at a given lag time τ is described by the correlation amplitude G(0), by diffusive processes GD(τ), by the photophysics GT(τ), and by the correlation offset Off. The density of protein can be obtained because G(0) is inversely related to the average number of particles in the confocal volume (G(0) ∝ N⁻¹). To obtain absolute density values the size of the confocal volume must be known. This is obtained from the calibration of free dye in solution. To obtain details on the confocal volume for 2D surfaces the following equation was used:

$$D = d^2 / [8 \cdot \ln(2) \cdot \tau D] \quad (2)$$

The equation can be rearranged to identify d, which represents the beam diameter or observation spot size. This is obtained from an assumption of a circular cross-section at the center and highest intensity region of a laser beam (i.e., a gaussian laser beam; one with a transverse intensity profile that approximates a gaussian distribution). D is the diffusion coefficient that can be obtained from the literature. The transit time τD is the average time taken for a dye to pass through the confocal volume. This is obtained from fitting an autocorrelation plot and taking the decay time. Once the diameter is obtained, the 2D area can be calculated, and therefore the density of labeled and mobile proteins on an SLB. The diameter (d) when using Alexa-647 dye at 37 °C was ~290 nm.

## Altering SLB composition

We used high levels of pMHCⁿᵘˡˡ to mimic the background of self pMHC presented on APCs but also to keep total protein density constant on the SLB. pMHCⁿᵘˡˡ was exchanged for other proteins (e.g., CD58, CD43, CD45, and ICAM-1) during production of the protein mix to modify SLBs. To ensure protein densities on the SLB were physiological they were matched to the mature monocyte-derived dendritic cell surface. To obtain the proteins within the density ranges consistent with moDCs, different combinations of protein concentrations were tested and analyzed using pFCS, using a trial-and-error approach wherein one protein was labeled with Alexa-647 and the others were unstained. To obtain the other protein densities for a given set of protein concentrations, the Alexa-647 labeled protein was switched to an unstained one at the same concentration, and a previously unstained protein was switched for an Alexa-647 labeled protein. For example, to obtain protein densities for SLB1s, three separate SLBs were produced to test one concentration set, whereas SLB2s required six. The final concentrations used for SLBs (in a 5 µl volume) are shown in Table 1 below.

## Altering agonist pMHC density

For pMHC density measurements and prior to placing cells on SLBs, Alexa-647 labeled pMHC⁹ᵛ on the SLB was subject to 3 × 10 s pFCS measurements using a 40x water objective (NA1.2; with 633 nm laser at 1% laser power). To alter the density of pMHC⁹ᵛ, ceteris paribus, it was exchanged directly for pMHCⁿᵘˡˡ; e.g., 0.5 ng/µl pMHC⁹ᵛ and 9.5 ng/µl pMHCⁿᵘˡˡ becomes 0.1 ng/µl pMHC⁹ᵛ and 9.9 ng/µl pMHCⁿᵘˡˡ. As the lower limit of detection for pFCS in this setup was ~1 molecule/µm², standard curves were produced to extrapolate densities below this. This was achieved by performing a four-point titration of density versus concentration. The three data points were measured above 1/µm² and were used to extrapolate the 4th lower density by assuming a linear relationship between density and concentration that passes through 0,0. The regression line equation could then be used to calculate concentration and density of agonist pMHC. For SLB1, the equation was Y (molecules/µm²) = 132.6*X(concentration in ng/µl). For SLB1 + glycocalyx, the equation was Y (molecules/µm²) = 327*X(concentration in ng/µl). For SLB1 + CD58, the equation was Y (molecules/µm²) = 434.8 × X(concentration in ng/µl). For SLB2, the equation was Y (molecules/µm²) = 185.5*X(concentration in ng/µl).

## Calcium release assay

Jurkat-derived cell lines or human primary CD8⁺ T cells that were primed and rested were either labeled with Fluo-4 or expressed the genetically encoded calcium sensor (GECI) jGCaMP7s. For Fluo-4, ~0.5 × 10⁶ cells were washed in PBS and placed in 1:1 mix of RPMI (no supplements). 25 µg/ml of Fluo-4 dye (final concentration; Thermo-Fisher) was added, and cells left for 15 min at 37 °C. Cells were washed in pre-warmed PBS plus 2 mM MgSO₄ and resuspended in the same. For the GECI, ~0.5 × 10⁶ cells were washed in pre-warmed PBS plus 2 mM MgSO₄, resuspended in the same. In all cases cells were incubated for a further 5 min at 37 °C. After the 5-min incubation, cells were gently placed onto the target surface and imaged using a 10x magnification objective for a larger field of view to simultaneously analyze 10²–10³ cells. Cells were imaged every 1 s. Both Fluo-4 and GECI were

**Table 1 | Protein concentrations used for different SLB compositions**

| Protein | SLB1 Conc. (ng/µl) | SLB1 + CD58 Conc. (ng/µl) | SLB1 + glycocalyx Conc. (ng/µl) | SLB2 Conc. (ng/µl) | SLB2/3x glycocalyx Conc. (ng/µl) |
|---|---|---|---|---|---|
| pMHC$^{9V}$ | 0 | 0 | 0 | 0 | 0.5 |
| pMHC$^{null}$ | 12 | 14.8 | 11.05 | 10 | 1.78 |
| ICAM-1 | 2 | 1.2 | 1.2 | 1.2 | 1.2 |
| CD58 | 0 | 0.5 | 0 | 0.5 | 0.375 |
| CD45 | 0 | 0 | 2.5 | 2.5 | 6.3 |
| CD43 | 0 | 0 | 1.5 | 1.5 | 4.5 |

excited using an argon 488 nm laser. Calcium imaging was performed on a Zeiss LSM 780 inverted confocal scanning microscope using a 10× objective. Cells were analyzed using a bespoke MATLAB code.

## Automated cell tracking and calcium analysis

To facilitate the analysis of calcium responses across hundreds to thousands of cells an automated MATLAB script was used (https://github.com/janehumphrey/calcium). The script tracks the movement and fluorescence intensity of the calcium dye (or any fluorescent label) for 100–1000 s of individual cells simultaneously. It first removes background, flat-field corrects, and gaussian smooths each video. Local maxima are then identified from the calcium signal, and the maxima from each frame are combined into contiguous tracks using a nearest-neighbor approach. The displacement and speed of each cell can be calculated. The time to adhesion is also calculated by a user-set threshold of speed, and when the cell speed drops and stays below this threshold for the remainder of its track a cell is classified as adhered. This was set at 0.2 µm/s based on the highly adherent interaction of Jurkats with unblocked glass. The mean intensity of the calcium signal is tracked over time for each cell. Time for 50% of all cells to adhere was extrapolated from fits of cumulative distribution plots of fraction of cells adhered, versus time for each SLB. Curves were fitted to the cumulative distribution plots constrained only to a maximum value of 100%, i.e., that all cells would eventually have a speed that dropped below 0.2 µm/s. Of note, displacement/speed is typically due to the currents generated even when gently adding cells to the SLB, which we exploited to address the adhesiveness of different SLB compositions.

The start of a calcium release (i.e., a proxy for TCR triggering events) was identified from a positive rate of increase in calcium signal from an identified background, calculated individually for each cell i.e., the lowest 10% of a cell's fluorescence intensity typically derived from the first 30–60 s of a cell track. To control for any variation in baseline fluorescence levels, thus allowing a comparison of calcium responses within and between samples, each calcium intensity trace was normalized by setting the baseline to one. User-set thresholds for identifying triggering events were also set, e.g., minimum duration and fold change from baseline for each increase in calcium signal. These criteria were set stringently with a minimum duration of 10 s and a fold change from baseline at 3 to remove false-positive increases in fluorescent signal derived from noise or cells settling on top of/near each other. These thresholds were set based on comparing the results of automated versus manual analysis of cell activation from several videos. Cells that showed calcium release within 40 s of the start of their measurement track were excluded from the analysis. These cells (typically 5–10% of all cells) settled and produced a bright calcium signal that then faded over the course of the video. This gave the appearance of an initial calcium spike to the code and was erroneously detected as an activated cell. These cells may represent pre-activated cells derived from charge-based interaction with Eppendorf tubes or

from pipetting. Time for 50% of all cells to signal was extrapolated from fits of cumulative distribution plots of the fraction of cells signaling, versus time for each SLB. Curves were fitted to the cumulative distribution plots constrained to a minimum of 0 and a maximum value of ~80%, based on the fraction of cells responding to OKT3 on glass on the day within a 10-min video; ~80% represented the upper limit of signaling given that most cells (>75% of all cells) would respond on OKT3-coated glass within 2 min. Overall, the code provides details on kinetics of movement, cell speed, adhesion, and cell activation as well as more in-depth details such as the strength of the calcium response, e.g., height, length, and integrated calcium increase. These metrics can be combined providing information about the time cells respond relative to settling, and the probability of a cell responding based on its time adhered.

## Calcium, IRM, and synapse imaging

Calcium release assays were performed as above. Immediately after, with the same SLB, and on the same microscope, the 10x objective was switched to a NA1.4 63x oil immersion objective to image cells in brightfield, the IRM contact area, and synapse formation. The confocal was set up to allow IRM images by choosing an appropriate emission filter that allows incident light reflected from the sample to be detected[73]. To visualize synapses, ICAM-1 was labeled with Alexa-555 prior to addition to the SLB. Cells were manually segmented and IRM contact area was quantified using Yen thresholding and the Analyze Particle feature in Fiji. Synapse formation was manually quantified using the fraction of all cells in a field of view that had formed an ICAM-1 ring.

## Scanning electron microscopy

13-mm diameter glass slides were incubated with poly L-lysine (PLL; 70–150 kDa MW; Sigma) according to the manufacturer's protocols. $1 \times 10^6$ J8 cells were fixed in 4% PFA 0.25% glutaraldehyde in PBS overnight at 4 °C. Fixed cells were gently placed on coated slides and left to settle for 20 min at room temperature. Slides were gently rinsed in PBS and fixed using 1% $OsO_4$ incubated at room temperature for 30 min. The sample was washed in PBS and dehydrated using increasing concentrations of ethanol (50 to 100% in 10% increments) over an hour. Ethanol was removed and 0.5 ml hexamethyldisilazane added for 3 min. This was removed and cells dried in a fume cupboard. Glass slides were mounted on carbon adhesive tape, sputter coated with gold, and viewed using a JEOL-6390 Scanning Electron Microscope.

## Confocal imaging of fixed T cell microvilli

$0.5–1 \times 10^6$ J8 cells were labeled with anti-CD62L (L-selectin) antibody labeled with Alexa-647 at 4 °C for 1 h. Cells were washed in PBS, then fixed in 4% PFA and 0.25% glutaraldehyde for 20 min at room temperature. During this time cells were labeled with 1x CellMask Green Plasma Membrane Stain (ThermoFisher). Cells were washed in PBS and immediately placed on PLL-coated glass surfaces (previously washed with ethanol) for 10 min prior to imaging. Imaging was performed on an LSM 880 scanning confocal with AiryScan within 30 min of plating. Cells were specifically imaged from the midplane to the apical surface to ensure microvilli in contact with the PLL (basal planes) were not included, in case the interaction altered protein organization even after fixation. Z-stacks were taken at 200 nm slices every 2 s. Cells were imaged using a NA1.4 63x oil immersion objective. Line scan averaging was set to a maximum of 8. Argon 488 nm, DPSS 561 nm, or He-Ne 633 nm laser power was set at appropriate levels to reduce bleaching and oversaturation. For measurements, the pinhole was set to 1 AU and a 488/594/633 MBS was used. Images were processed using Fiji with the Z-projection (maximum intensity) feature. Fifteen frames at 200 nm spacing were used per cell analyzed. The microvillus diameters were quantified by manually choosing membrane protrusions

with clear L-selectin enrichment, obtaining the cross-section line profile, and using the full width at half maximum of the increased signal as a measure of the diameter.

## Actin-drug modification of T-cell morphology (fixed-cell imaging)

$0.5 \times 10^6$ J8-GECI cells were washed ×1 in PBS and resuspended in either 10 μM DMSO (control), 1 μM latrunculin B, and 10 μM cytochalasin D, or 100 nM jasplakinolide (Cayman Chemical) diluted in RPMI (no supplements) for 1 h at 37 °C with 5% $CO_2$. Cells were then fixed in PBS containing 4% PFA and 0.25% glutaraldehyde for 20 min at room temperature. During this period the cells were also labeled with Cell-Mask Deep Red Plasma Membrane Stain (ThermoFisher). Cells were washed ×2 in PBS and immediately placed on PLL-coated glass surfaces (previously washed with ethanol) for 10 min prior to imaging at room temperature. Images were taken as described for imaging of fixed T cell microvilli.

## Actin-drug modification of T-cell morphology (live-cell imaging)

During drug treatment of the cells, SLBs were made containing either no glycocalyx elements (i.e., SLB1 + CD58; pMHC^null, pMHC^9V, ICAM-1, and CD58) or with the glycocalyx (i.e., SLB2; pMHC^null, pMHC^9V, ICAM-1, CD58, CD45, and CD43). The cells and SLBs were washed and resuspended in pre-warmed PBS plus 2 mM $MgSO_4$ with the same concentration of actin-modifying drug as above, and incubated at 37 °C for 5 min prior to use. Calcium release was imaged using a 780 LSM scanning confocal for 10 min. To relate drug treatment to contact formation, prior to dropping cells on SLB2s, the cells were labeled with CellMask Deep Red. J8-GECI cells were labeled using 1x CellMask Deep Red for the final 10 min of actin-modifying drug treatment. CD45 and CD43 were labeled with Alexa-555 to observe close contacts. Cells were then washed and resuspended in pre-warmed PBS plus 2 mM $MgSO_4$. SLBs and membrane-labeled cells were incubated at 37 °C for 5 min prior to being gently deposited onto the SLBs and immediately imaged by confocal microscopy. Calcium release, cell membrane, and close-contact formation were imaged every 10 s for 10 min. Confocal videos were analyzed using our custom contact analysis Python script given below, with user-input parameters modified to match the metadata from the confocal microscopes.

## TIRF microscopy

Three-color live cell imaging of SLB-cell contacts was performed on a bespoke TIRF microscope. The excitation path comprised laser lines of 488 nm (iBeam-SMART, Toptica), 561 nm (LaserBoxx, DPSS, Oxxius), and 641 nm (Obis, Coherent). Each beam was circularly polarized using quarter-wave plates, collimated, and expanded. Laser lines were combined using appropriate dichroic mirrors and expanded further to reduce the flat-field variation. The lasers were aligned off-axis at the edge of the objective lens (100x Plan Apo TIRF, NA 1.49 oil-immersion, Nikon Corporation) and focused onto the backfocal plane to achieve TIR illumination. The objective was mounted on an inverted optical microscope (Ti2, Eclipse, Nikon Corporation). The fluorescence emission was collected through the same objective and separated from the excitation light via a dichroic mirror (Di01-R405/488/561/635, Semrock). It was further passed through appropriate filters according to the excitation wavelength (FF01-520/44-25 + BLP01-488R, LP02-568RS-25 + FF01-587/35-25, FF01-692/40-25; Semrock), mounted in an automated filter wheel. The fluorescence was expanded (1.5x) and the image formed onto an electron-multiplying charge-coupled device (EMCCD, Evolve 512 Delta, Photometrics) with an electron multiplication gain of 250 ADU/photon operating in frame transfer mode. This resulted in an effective pixel size of 107 nm. To allow for live-cell imaging the microscope was enclosed with an incubator (DigitalPixel) and the temperature maintained at 37 °C. The focus was maintained with the Nikon Perfect Focus System. Image acquisition was automated using the open-source software Micro-Manager. Three-color images were acquired sequentially at 0.5–2 Hz with 100 ms exposure time. The laser powers were optimized to reduce bleaching while maximizing signal.

## TIRFM of calcium signaling, cell footprint, and close-contact formation

SLB2s presenting the background of pMHC^null and the indicated density of agonist pMHC were prepared as described above. To visualize contact structure, CD45 and CD43 were labeled with Alexa-555. All other proteins on the SLBs were unlabeled. SLBs were washed and prepared in pre-warmed PBS plus 2 mM $MgSO_4$. To visualize the membrane of J8-GECI cells or primary CD8^+ cells, the cells were labeled using 10 μM CellMask Deep Red for 10 min in RPMI (no supplements) at 37 °C. To visualize calcium release, primary cells were labeled with Fluo-4. J8-GECI or primary cells were then washed and resuspended in pre-warmed PBS plus 2 mM $MgSO_4$. SLBs and cells were incubated at 37 °C for 5 min prior to being gently deposited onto the SLBs and immediately imaged by TIRFM. The SLB was imaged in the 5 min prior to depositing the cells in order to gain the background fluorescence for flat-field correction. A complete frame cycle was taken every 2 s for 10–20 min. Individual microvilli tips, observed as membrane puncta moving in and out of an evanescent field (as seen in Supplementary Fig. 2c), were analyzed using the Trackmate plugin in Fiji. The persistence time was defined as the time each membrane puncta/microvillus tip could be tracked before the fluorescent signal was lost. Only puncta visible prior to a cell settling on the surface were analyzed, i.e., prior to a close contact being formed.

## TIRFM of protein accumulation and exclusion at close contacts

SLB2s with Alexa-555 labeled CD45 and CD43 were prepared as described above. To image the TCR relative to close contacts, J8-GECI cells were incubated with 50 μg/ml of UCHT-1 Fab labeled with Alexa-488 for 10 min in RPMI (no supplements) at 37 °C. To image pMHC relative to close contacts (in a separate experiment), SLB2s were prepared using ~100 molecules/μm² of pMHC^9V-Alexa-647. To image ZAP70 (in a separate experiment), cells were labeled and prepared with 1x Janelia-Fluor 646 HaloTag Ligand in RPMI according to the manufacturer's instructions. To image L-selectin (CD62L) relative to close contacts (in a separate experiment), cells were labeled with Alexa-647 tagged L-selectin antibody for 10 min in RPMI (no supplements) at 37 °C. After washing both the cells and SLBs in pre-warmed PBS plus 2 mM $MgSO_4$, SLBs and membrane-labeled cells were incubated at 37 °C for 5 min prior to being gently deposited onto the SLBs and immediately imaged using TIRFM. The SLB was imaged in the 5 min prior to depositing the cells to obtain the background fluorescence for flat-field correction. A complete frame cycle was taken every 2 s for 10–20 min. For the enrichment analysis several FOV were taken afterward.

In analogous experiments, CD58 and ICAM-1 on the SLB2s were labeled with Alexa-647 to image their reorganization in relation to close contacts (in separate experiments). ICAM-1 was also labeled with Alexa-555 to be imaged alongside CD58 labeled with Alexa-647. Similarly, cells were labeled with ~15 μg/ml of anti-CD45 (Gap8.3 clone) tagged with silicon rhodamine (SiR) to label cell-expressed CD45. The use of the Gap8.3 Fab was impractical owing to its fast off-rate at 37 °C.

## Simulation of close-contact formation on supported lipid bilayers

Exclusion regions of varying extent of exclusion and size were simulated using Python and assuming a homogeneous intensity as ground truth. The ground truth SLB was an image with 1 nm pixel size and intensity 100 in which circular contacts with intensity values between 0 and 100 were added. The image formation was approximated by a Gaussian blur with sigma of the experimental point spread function of

the bespoke TIRF microscope and pixels binned to 107 nm size. The experimental point spread function had been measured using Tetra-Speck Microspheres and the sigma of the Gaussian approximation determined as 131 nm. Image brightness was adjusted to match experimental values and the EMCCD simulated using a noise model previously published[74]. The simulated images were segmented the same way as experimental TIRFM data to determine the smallest and least excluded contacts detectable.

## Analysis of close contacts

Quantitative image analysis of close contacts was performed with custom-written software in Python (https://www.python.org/) using the packages NumPy[75], scikit-image[76], scipy[77], and pandas[78], as well as matplotlib[79] and seaborn[80] for visualization. The software was divided into three parts: image segmentation, feature extraction and analysis, and advanced feature analysis for plotting. The full code is accessible at https://github.com/mkoerbel/contactanalysis_2D.

## Image segmentation

Three-color confocal or TIRFM time-lapse images (with dimensions (x, y, t, 3), acquired as described in "Actin-drug modification of T-cell morphology (live cell imaging)" or "TIRFM of calcium signaling, cell footprint and close-contact formation") were used to analyze contact dynamics of SLB-interacting T cells. The 641-excitation channel showed the T cell membrane, the 561 channel a protein in the SLB indicating close contacts (glycocalyx, CD43 and CD45), and the 488 channel the intracellular calcium levels of the T cell. First, the cell membrane was used to identify a cell. Because the membrane dye over time also labels the SLB itself, a linearly increasing background was assumed and corrected for. A framewise Difference of Gaussian (DoG) filter was then applied to each frame of this channel to reduce noise and an inhomogeneous background. A binary mask of the cell membrane was created by combining a global, custom-set threshold with a threshold calculated by the Otsu method, which allowed a better segmentation of large, spread cells. Masks with an area below a threshold were removed. Assignment of cell labels to each mask was done with the Watershed algorithm including the time dimensions (i.e., in three dimensions). T cells that formed contacts with the SLB showed reduced lateral mobility, maintaining their position in (x) and (y). Masks of adhered cells thus overlapped in consecutive frames. Using a larger DoG filter on the last cell membrane frame, cell positions were defined by the detected local maxima, separated by at least the cell radius, and used as seeds for the Watershed segmentation. These seeds could be manually edited, and seed-regions added in case the automatic labeling could not distinguish between closely associated cells. Disconnected regions were assigned to the closest labeled region, not further than twice the cell radius measured as Euclidean distance in (x,y,t), and tracing backward in time.

For the analysis of close contacts based on glycocalyx exclusion, TIRFM images were first divided by the flat-field. The flat-field was obtained by summing and normalizing a separate image stack, taken before T cells were added to the SLB. A rolling-ball filter was applied in (t), followed by a Gaussian filter in (x,y) on the corrected image stack. The Laplacian of the filtered image was calculated (giving a Laplacian of Gaussian, LoG, filter overall). Two thresholds were applied to account for the variability of contact size across different conditions: to segment close contacts a hysteresis threshold was used. The upper and lower boundaries were defined as $h \cdot \text{std}(I(\text{LoG})_{\text{notCZ}})$, with $h$ being two user defined inputs and $I(\text{LoG})_{\text{notCZ}}$ all pixel intensities of the calculated LoG outside of contacts (segmented membrane areas). To segment late, unconstrained contacts an edge enhanced image (sum of the Gaussian filtered image and the LoG filtered image) was thresholded via $\text{mean}(I(\text{Gauss})_{\text{notCZ}}) - h_2 \cdot \text{std}(I(\text{Gauss})_{\text{notCZ}})$ and morphologically eroded. Close contacts larger than a minimum size were labeled based on the detected cell labels.

## Feature extraction and analysis

To analyze the segmented images and extract contact features, the calcium response for each cell was first measured. The mean fluorescence intensity in a circle centered at the contact centroid was calculated for each timepoint. If a contact was missing at a given timepoint, it was linearly interpolated from the next neighboring centroids for that cell. The calcium trace was analyzed the same way as in the bulk calcium assay to obtain the timepoints of calcium triggering and adhesion. A quality control step was included to exclude cells that touched the edges of the FOV in the cell membrane channel either before they trigger or, if they do not trigger, within a given time period after the first close contact has been detected. Only cells passing this step were further analyzed. Several events that characterize the interaction of the SLB were defined at the cell level (see also the four interaction stages in the main text, and Table 2 for definitions of the interaction stages). Features for contacts and close contacts were based on area and intensity values and were calculated for each timepoint (see Table 3 for contact features and descriptors).

## Derived feature analysis

The total cell membrane time trace was smoothed using a mean filter and the maximum detected. Using this, and in the previous step defined event, each timepoint was assigned one of the four interaction stages of the cell. A summary of all calcium traces and total cell membrane areas was produced.

## Sensitivity and resolution limit

The sensitivity to detect close contacts with the presented close-contact analysis was evaluated by simulations (Supplementary Fig. 6c). Simulations showed that contacts with sizes below the diffraction limit could be detected, especially if the exclusion was higher than 40%. The filtering applied (temporal filtering and minimal size threshold) ensured that the segmentation of a small number of pixels was not due to noise.

The resolution limit was defined by the optical system, in our case an objective with NA 1.49. Diffraction caused small contacts to appear larger than their actual size. This effect was convoluted with the exclusion at that contact. Simple thresholding of a less excluded contact would result in a smaller contact. This effect was minimized by choosing an edge-based filter (LoG). Segmentation was done on a pixel basis, thus the smallest detectable contact has the area of 1 pixel (for 107 nm pixel size that area was 0.011 $\mu m^2$) which results in the reporting of close-contact areas below the diffraction limit. Any resulting bias would affect all data equally and not change the presented comparative differences or trends in our data.

## Cumulative density plots for different stages (searching to scanning, scanning to spreading)

Cumulative density function for the transition of each cell between stages was estimated using the Kaplan–Meier estimate for the survival function provided by the lifelines package in Python. The transition to the next stage was considered a "death" event. Median survival times were compared using the log-rank test.

## Detection of single-bound pMHC[9V] at close contacts

SLB2s with Alexa-555 labeled CD45 and CD43 and Alexa-647 labeled pMHC[9V] at a concentration of 1.25 pg/µl to achieve single-molecule densities, were prepared as described above. To identify engaged single pMHC[9V], the exposure time to image pMHC was set to 200 ms to achieve motion-blur for unbound molecules. Initially 5 frames of the whole FOV were taken without cells for flat-field correction of the CD45/CD43 channel and to estimate the analysis threshold. For analysis, a rolling-ball average of 5 frames was calculated and each frame Gaussian blurred with a sigma of 1 pixel. The local maxima above a set threshold were counted to report the number of bound pMHC per timepoint. The threshold was set so that fewer than 2 pMHC were detected in the initial

**Table 2 | Interaction events**

| Event | Output string | Description |
|---|---|---|
| Cell membrane detected | time_CZ_first (s) | Time when the first cell membrane signal was segmented. Initiation of "searching" stage. |
| Cell adhesion | time_adhesion (s) | Time when cell speed first falls below threshold. |
| Close-contact formation | time_CCZ_first (s) | Time when the first close contact was detected. Initiation of "scanning" stage. |
| Calcium signaling | time_Ca (s) | Time when the cell triggers. Initiation of "spreading" stage. |
| Maximum cell membrane area | time_to_CZ_max (s) | Time relative to maximum detected cell membrane area. Initiation of "synapsing" stage. |

**Table 3 | Contact features and descriptors**

| Feature | Output string | Description |
|---|---|---|
| Area | Area ($\mu m^2$) | The area of a contact based on the number of segmented pixels. |
| Exclusion | exclusion_10 | Only applicable to close contacts. The exclusion of glycocalyx from close contacts, defined as $1 - \mathrm{perc}(I_{inside})/\mathrm{mean}(I_{outside})$, with $\mathrm{perc}(I_{inside})$ being the 10th percentile of intensity values inside the close contact, and $\mathrm{mean}(I_{outside})$ being the mean of intensities on the 1 pixels wide outline of the segmented cell membrane. Intensity values were taken from the flat-field corrected images. |
| Contact time | contact_time (s) | Time the individual close contact has been detected for. In order to be "tracked" from frame to frame, the segmented areas need to overlap temporally. "time_evol" classifies the end and start of the contact. |
| Stage | stage | For a contact at a given timepoint, in which stage of the interaction the cell currently is. |

image stack. The timepoint of signaling was determined as the maximum in the calcium signal gradient and cells aligned with it. Close contacts were segmented using a rolling-ball average, LoG filter, and hysteresis threshold as described above. If the detected local maximum was within the segmented close-contact area, it counted towards the number of bound pMHC within that contact.

**Enrichment analysis of proteins on SLB2s**
The enrichment of proteins relative to close contacts was calculated from TIRFM images with at least one channel devoted to the glycocalyx in the SLB. The glycocalyx channel was flat-field corrected and segmented with the same double-threshold approach as in 'Analysis of close contacts.' The thresholds were adjusted for each set of images from the same SLB. This defined regions "inside" the close contacts. Each region was morphologically diluted by 10 pixels. In this way, obtained "outside" regions were further refined by excluding parts that overlapped with another "inside" region. If the protein of interest was on the cell, the cell was segmented and "outside" regions that were not within the segmented cell region were excluded as well to avoid an apparent reduction in signal at the cell membrane edge. To calculate the enrichment of the protein of interest, which had been imaged in another color channel, the mean intensity of the protein signal from the "inside" regions, $I_{inside}$, and "outside" regions, $I_{outside}$, for each close contact, was calculated. The enrichment was plotted for each close contact as $\mathrm{mean}(I_{inside})/\mathrm{mean}(I_{outside})$.

**Close-contact area as a fraction of total membrane area at signaling**
Total membrane area was quantified for J8-GECI cells and primary CD8$^+$ T cells in the same manner for moDCs, as described in the methods section titled 'Protein densities on moDC'. The surface area was ~525 $\mu m^2$ for J8-GECI and ~300 $\mu m^2$ for primary T cells. Total close-contact area at signaling was divided by the total surface area of a cell in solution to provide the close-contact area as a fraction of total membrane area at signaling.

**Analysis of CD45 exclusion at close contacts**
The TIRFM videos obtained were first analyzed as described in "Analysis of close contacts", with the CD45 channel used as a cell membrane stain, adjusting segmentation accordingly, and not restricting the segmented close contacts to segmented cell membrane areas. CD45 exclusion was further analyzed by first obtaining "inside" and "outside" regions as defined in "Enrichment analysis of proteins on SLB2s" using the already segmented close contacts. The exclusion was calculated as

$1 - \mathrm{perc}(I_{inside})/\mathrm{mean}(I_{outside})$, with $1 - \mathrm{perc}(I_{inside})$ being the 10th percentile of intensity values inside the close contact.

**Tracking CD58 accumulation under the TCRKO-GECI cells**
Individual sites of CD58 accumulation (a proxy for close contacts) were analyzed using the Trackmate plugin in Fiji.

**Image display**
Imaging data were visualized using Fiji (https://imagej.net/Fiji[81]). All images within a figure panel were contrast matched. TIRFM images of the glycocalyx in the bilayer were flat-field corrected as described in 'Analysis of close contacts.' For proteins that accumulate under cells on the bilayer (i.e., CD58, ICAM-1, pMHC), and for image presentation purposes, background fluorescence was subtracted across the whole video. Background fluorescence was removed using a z-projection (average) of a 60 s video (taken at 0.5 Hz) of the bilayer taken prior to adding cells. All TIRFM images of the cell or bilayer proteins were z-projected (average) across 3 frames giving a temporal resolution of 6 s.

**Cartoon representations of proteins attached to the SLBs**
Protein structure data were obtained from the Protein Data Bank (https://www.rcsb.org/)[82]: 2BNQ (1G4-TCR in complex with pMHC[40]), 1HNF (CD2)[83], 5FMV (CD45)[49], 1CCZ (CD58)[84], 1Z7Z (ICAM-1)[85], 5ES4 (LFA-1)[86], 5T78 (MUC1 repeated for CD43)[87]. The structures were rendered using the online "Illustrate" tool (https://ccsb.scripps.edu/illustrate/)[88] and assembled in Adobe Illustrator. The CD2/CD58 complex was based on the structure of the complex of their N-terminal domains (1QA9)[89], and superpositions of the structure of the full-length ECD of CD2 (1HNF).

**Statistical analysis**
Statistical significance was calculated for all quantitative data using GraphPad Prism or the Python programming language and statistical tool packages scipy and scikit-posthocs[90]. The number of cells per experiment, the number of experiments per measurement, the statistical significance of the measurement, and the statistical test used to determine the significance are indicated in each figure legend where quantification is reported. In general, significance was defined based on either Student's t-test (two-sided), for comparing replicates, or Mann–Whitney U test, for comparing individual cells, computed on the mean or mean rank values from independent experimental replicates or one-way ANOVA with an appropriate multiple comparisons test. In most cases $p$ values were reported in the figure. For comparing

contact features across different conditions either the $p$ values of the Kruskal–Wallis H test are reported or, if there was a significant difference in the group ($p < 0.05$), individual $p$ values of the pairwise Mann–Whitney U test are reported if they were below 0.05.

## Reporting summary

Further information on research design is available in the Nature Portfolio Reporting Summary linked to this article.

## Data availability

Most of the raw TIRF imaging data can be accessed at 10.5281/zenodo.7509910. Sample data for cell and calcium tracking using the confocal microscope can be accessed at 10.5281/zenodo.7510290. All additional datasets referred to in the current study are available from the corresponding authors on request. Source data are provided in the Source data file. Source data are provided with this paper.

## Code availability

TIRFM-based close-contact analysis code is available at the Github repository mkoerbel/contactanalysis_2D, 10.5281/zenodo.7517858, 2023 (https://github.com/mkoerbel/contactanalysis_2D). Calcium signaling analysis code is available at the Github repository janehumphrey/calcium, 10.5281/zenodo.6390856, 2022 (https://github.com/janehumphrey/calcium).

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

## Acknowledgements

The authors thank Professors P. A. van der Merwe, P. Jönsson, M. L. Dustin, and Dr. Y. Lui for helpful discussion and comments on the manuscript, Professor O. Dushek and Dr. J. Pettman for providing sequences, prior to publication, of weakly-agonistic peptides, A. Mørch for providing the pHR-Lck-HaloTag plasmid, M. Vuong for pMHC refolding, the staff of the WIMM FACS Facility for cell sorting, and Dr. C. Lagerholm for help with the microscopy. We also thank the staff of the Wolfson Imaging Centre, University of Oxford, for providing access to their imaging facility. This work was funded by the Wellcome Trust (Grant 207547/Z/17/Z awarded to S.J.D.) and by a Royal Society Research Professorship (RP150066 awarded to D.K.).

## Author contributions

E.J., M.Kör., S.J.D., and D.K. conceived and designed the study; E.J., M.Kot, A.M.S., and H.B. produced cell lines and reagents; E.J., M.Kör., C.O'B.-B., M.Kot, J.M., K.C., and A.L. performed experiments; E.J. and M.Kör. analyzed the data; M.Kör., S.L., and J.H. created analysis codes; E.J. and S.J.D. wrote the paper. E.J., M.Kör., S.J.D., and D.K. edited the manuscript; S.J.D. and D.K. oversaw the study.

## Competing interests

The authors declare no competing interests.
