## [Peer Review File · Nature Communications]

Antigen discrimination by T cells relies on size-constrained microvillar contactREVIEWER COMMENTS

Reviewer #1 (Remarks to the Author):

The role of microvilli in mediating the interaction of T lymphocytes with other cells in the early stages of the immune response has become clearer in recent years. Novel microscopy techniques have shed light both on the organization of key proteins with respect to microvilli, but also on their dynamics during interaction with target cells.

This exciting paper adds a new dimension to studies on the role of microvilli. Realizing that target cells are often covered with a glycocalyx layer, the authors aim at building a glycocalyx-mimetic system that would allow them to study the interaction of this layer with microvilli and the proteins residing on them, and to gauge the importance of such interactions for the activity of the T cells. Using an impressive combination of techniques and a very broad arsenal of molecular tools, the authors manage to demonstrate mutual interaction between glycocalyx proteins, adhesion molecules and signaling molecules.

Before the paper is accepted for publication in Nature Communications, the authors should deal with the following issues:

Major points:

1. After introducing the two glycocalyx proteins used in their study, the authors continue describing them as 'the glycocalyx'. However, in reality these two proteins are just a small part of the complete glycocalyx, which is a 50-100 nm layer of glycolipids and glycoproteins. The potential role of microvilli becomes clear when one thinks about the necessity to penetrate this thick layer, but with only two proteins on the surface, the whole penetration process seems redundant and unrequired. This issue should be clarified, and the authors should refrain from calling the two proteins 'the glycocalyx', and rather find a more 'modest' name. In this context, it seems that the discussion on the size of interaction/contact areas needs to be modified, as the way microvilli would interact with a 'real' thick glycocalyx would be quite different from the way observed in this work. One related question is whether the two glycocalyx proteins are sialylated, as they apparently should be.
2. The authors often observe black holes forming on the image plane when cells are interacting with the surface, and interpret these holes as areas depleted of the glycocalyx proteins. However, no proof is provided for such a depletion. The black holes could be, for example, areas where fluorescence is simply quenched. One way to prove protein depletion is to repeat the experiments with the proteins immobilized on a non-SLB (glass?) surface and show that in that case black holes are not forming.
3. It is unclear how CD2 is involved in the processes discussed here- is there evidence that it is enriched on microvilli, or is it distributed over the whole cell body? This should be discussed and potentially experiments to shed light on this issue should be carried out.
4. There is a lot of discussion of the size of features in images. It seems that the diffraction-limited nature of confocal microscopy, which prevents resolving features much smaller than the wavelength, is ignored here, especially when features of less than 0.5 micron are discussed. How does the diffraction limit affect all the size analyses presented in the paper?

Minor points:

1. Page 2- the statement on the time it takes to exclude CD45 molecules is not clear and its context is hard to follow.
2. Page 3- the acronym SLB should be defined the first time it appears.
3. Page 4- please define the specificity of "1G4 TCR".
4. Page 5- it is not clear what the dwell-time kinetics discussed here are and how they are quantified.
5. Page 6- please provide a general explanation on how close contacts are quantified.
6. Same page- the different treatments affect microvilli in different ways, some paralyzing them, some probably collapsing them. These differences and their effects should be clarified.
7. Page 8- It requires more than a single movie to convince that 1-2 pMHC molecules are involved in the contacts observed here. More data and quantification are needed.
8. Same page- "suggesting that CD2 enhance pMHC detection..."- the way this conclusion is reached is not clear.
9. Same page- ICAM-1 accumulation in rings- but the features observed don't look like rings at all.... There's a diffuse fluorescent region with a hole in the center, but why call this a ring?

10. Page 9- "In the absence of CD58 cells were also more mobile"- this requires a better explanation. For example, what the displacement tracks in Figure 4j are is not very clear.
11. Page 10- the exclusion of CD45 from areas of CD58 accumulation is discussed. In this context, it has recently been shown the CD45 is completely excluded from microvilli tips (Jung et al., <https://doi.org/10.1038/s41467-021-23792-8>). How does this finding reflect on the way the authors discuss here the role of CD45?
12. Page 20: when multiple measurements are taken, is the focus stable enough and doesn't it drift?
13. Same page: can equations be written more appropriately, using e.g. subscript and superscript letters when necessary?

Reviewer #2 (Remarks to the Author):

In this interesting study, the authors expand the repertoire of tools for studying early T cell:APC engagement, signaling and synapse formation. Thus, they start with the now-standard supported lipid bilayer (SLB), which includes ICAM-1 and peptide-MHC, and add to it CD58 (a ligand for CD2), and CD43 and CD45 (large glycosylated proteins that are a key part of the "glycocalyx"). Using this more physiologically representative SLB (dubbed "SLB2"), they show that the more complex SLB setup allows for a fine T cell response that balances sensitivity and specificity. Interestingly, including only CD58 or CD43/CD45 in the SLB1 setup leads to loss of this balanced response. This led to the authors to further investigate the reasons for the difference in these divergent responses. In brief, they find that the earliest, probing, interactions of T cells and the SLB's are mediated by microvilli, which can penetrate the glycocalyx. These contacts are further regulated by CD2:CD58 interactions, which limit the size of the contacts. Together, this study provides significant insight into previous observations that demonstrated the presence and dynamic nature of T cell microvilli. It also provides strong support for future use of the SLB2 arrangement as a more biologically relevant model for studying the earliest events in T cell activation.

Overall, this is clearly presented manuscript, including the writing, figures and supplemental movies, the latter of which nicely extend on findings shown in the figures. I have a few comments and suggestions for the authors to address:

1. I like the inclusion of the Ca²⁺ signaling reporter, which provides a rapid readout for early signaling. However, I do not think it is fair to use it as a shorthand for all signaling, which the authors seem to imply in some places (although this may not have been their intention). Calcium signaling is quite dynamic in T cells, so early bursts of free Ca²⁺ may not fully represent later signaling events, including by other pathways (e.g. Ras/MAPK).
2. On a related note, have the authors examined whether free Ca²⁺ is necessary for downstream events examined in this system? Thus, would Ca²⁺ chelation still allow the early contacts but prevent spreading? Similarly, if Ca²⁺ were chelated immediately after the initial signaling event, would that prevent the spreading and synapsing?
3. The authors extend their studies by including some experiments with primary CD8⁺ T cells. I realize that these experiments have some challenges, but I am not completely convinced that the UCHT1 Fab is a fair comparison in all cases. The authors should discuss this.

Reviewer #3 (Remarks to the Author):

T cells use finger-like protrusions called 'microvilli' to scan the surface of antigen-presenting cells. The present paper provides a rationale for such strategy. Microvilli have been implicated in force driven penetration of the glycocalyx and they use relatively 'small' adhesive proteins (primarily CD2, a receptor for CD58 that is also endowed with stimulatory properties) to form close contact zone that are thought to be the site of initial TCR triggering. The authors recently showed that artificial and natural TCR ligands enhance receptor phosphorylation and signaling by 'trapping' TCRs in such

phosphatase-depleted close contacts. Here, the authors experimentally demonstrated a hypothesis they published in 2019 (see Ref. 61) and that posited that the small and uniform cellular contacts formed by the microvilli upon penetrating the glycocalyx are essential to equip T cells with the capacity for self vs non-self discrimination by T cells.

The authors developed (1) modified SLBs presenting physiological levels of 'null' pMHC, agonist pMHC (pMHC9V), ICAM-1, CD58, CD43 and CD45, and (2) a human CD4-CD8 $\alpha\beta$ + 1G4 TCR-expressing Jurkat T-cell line, expressing additional LFA-1 molecules to match those of primary CD8+ effector T-cells, and a genetically encoded calcium indicator. 'Second generation' SLB (SLB2) presenting CD58 and a glycocalyx together with pMHC and ICAM-1 allowed the interplay between the glycocalyx and adhesion proteins to be investigated. They demonstrated that the glycocalyx alone substantially suppressed signalling and that together with adhesive proteins it enforced the specificity and sensitivity of antigen recognition by 1G4 T cells. Using labeled CD43 and CD45 molecules inserted onto SLB2 and TIRF microscopy, 'holes' were observed to form in SLB2 corresponding to local exclusion of glycocalyx elements and corresponding to regions of heightened TR density and ZAP-70 fluorescence. These close contacts were reduced upon treatment with actin-modifying drugs. Using a three-color TIRF analysis gauging calcium signalling, interaction footprint, and exclusion of glycocalyx components on the SLB2 presenting pMHCnull and two levels of pMHC9V ligands, four distinct stages of contact were identified and denoted as 'scanning', 'searching', 'spreading', and 'synapsing'. The transition from scanning to searching was independent of pMHC, whereas the progression to spreading was pMHC dependent. Moreover, contacts formed prior to calcium signaling were highly constrained in size. Measurements of contact area at onset of calcium release indicated that pMHC sensing was highly efficient, requiring <0.2% of the total cell surface area. Using single particle tracking on SLBs presenting pMHC9V-lo, 1-2 pMHC9V were found to be engaged at close contacts at onset of calcium release. The TCR was found dispensable for close contact formation and stabilization. In contrast, CD2 enhanced close contact formation and stabilization, whereas ICAM-1/LFA-1 interactions control the cellular footprint that follows calcium signaling. CD2-deficient J8-GECI cell line (CD2KO) were also developed and reconstituted with wild-type CD2 (CD2WT) or CD2 whose cytoplasmic domain was deleted (CD2 Δ CYT). CD2WT and CD2 Δ CYT excluded ~40% and ~32% of CD45 from close contacts, respectively. In marked contrast, CD2KO cells formed fewer and smaller close contacts. Unexpectedly, CD45 exclusion at each CD2KO contact was similar to CD2 Δ CYT.

To examine the effects of increasing contact size on pMHC discrimination, the authors capitalized on J8-GECI T cells that over-express CD2 by a factor of 10 and form 5- to 10-fold larger close contacts than CD2WT-expressing J8-GECI T cells. CD2KO, CD2WT, and CD2WThi cells on pMHC null-presenting SLB2s produced calcium responses of 5%, 20%, and 65%, respectively, revealing a loss of antigenic discrimination as contact size increases. Finally, in contrast to the J8-GECI cells expressing native levels of CD2, approximately 70% of CD2WThi cells produced calcium responses irrespective of pMHC affinity.

The results of this comprehensive, solid, and elegant study that provides a dual 'raison d'être' for the use of microvilli by T cells are fairly discussed.

Minor comments:

1/ The meaning of SLB needs to be specified.

2/ Have the authors an explanation for the fact that CD45 exclusion at each CD2KO contact was similar to CD2 Δ CYT?

“The central role of microvilli in shaping T-cell responses”

Jenkins et al.

Reviewer #1 (Remarks to the Author):

The role of microvilli in mediating the interaction of T lymphocytes with other cells in the early stages of the immune response has become clearer in recent years. Novel microscopy techniques have shed light both on the organization of key proteins with respect to microvilli, but also on their dynamics during interaction with target cells.

This exciting paper adds a new dimension to studies on the role of microvilli. Realizing that target cells are often covered with a glycocalyx layer, the authors aim at building a glycocalyx-mimetic system that would allow them to study the interaction of this layer with microvilli and the proteins residing on them, and to gauge the importance of such interactions for the activity of the T cells. Using an impressive combination of techniques and a very broad arsenal of molecular tools, the authors manage to demonstrate mutual interaction between glycocalyx proteins, adhesion molecules and signaling molecules.

Before the paper is accepted for publication in Nature Communications, the authors should deal with the following issues:

Response to reviewer #1

We thank the reviewer for their generous and very fair analysis of our study.

Major points:

1. After introducing the two glycocalyx proteins used in their study, the authors continue describing them as ‘the glycocalyx’. However, in reality these two proteins are just a small part of the complete glycocalyx, which is a 50-100 nm layer of glycolipids and glycoproteins.

We wish to emphasize that our SLB2 bilayers seek to model the APC bilayer, which with respect to its glycoprotein composition comprises principally of CD43 and CD45, the two largest, most abundant proteins expressed by leucocytes. We accept, however, that the use of ‘the’ glycocalyx may be misleading given this constraint. We have now referred more explicitly to the APC glycocalyx in establishing the SLB2 model and, elsewhere, changed “the glycocalyx” to either “a glycocalyx”, “model glycocalyx”, “glycocalyx elements” or “glycocalyx barrier”, as appropriate. We also de-emphasized the glycocalyx in the introduction in favour of microvilli, given that we are restricting ourselves to one example of a glycocalyx, and also because it is really the role of microvilli that is, we think, the most important aspect of our study.

The potential role of microvilli becomes clear when one thinks about the necessity to penetrate this thick layer, but with only two proteins on the surface, the whole penetration process seems redundant and unrequired.

We agree with the reviewer that in the presence of a more complex glycocalyx there is likely an even greater need for microvilli to generate close contacts. However, we do not agree that the use of microvilli to penetrate our SLB glycocalyx is unnecessary in our system. After all, adding the model glycocalyx to the SLB1 substantially reduced calcium responses at all pMHC9V levels (by >30%) and introduced delays in the response (up to 2-3-fold; Fig. 1c).

This issue should be clarified, and the authors should refrain from calling the two proteins ‘the glycocalyx’, and rather find a more ‘modest’ name.

As indicated above, we have now changed “the glycocalyx” to, e.g., “a glycocalyx”, “model glycocalyx”, “glycocalyx elements” or “glycocalyx barrier”, throughout the manuscript.

In this context, it seems that the discussion on the size of interaction/contact areas needs to be modified, as the way microvilli would interact with a ‘real’ thick glycocalyx would be quite different from the way observed in this work.

It is not immediately clear, *a priori*, whether the size of contact interaction/areas would be substantially different from what we observed for different glycocalyxes. We suspect that the lower limit of close contact size is established by the diameter of microvillar tips themselves. However, a “thicker” (deeper?) glycocalyx might reduce the frequency of penetrating the glycocalyx, slowing close contact formation (and subsequent antigen recognition).

Although we cannot directly address this in the present experiments, i.e., whether a thicker glycocalyx would impact contact size, in response to the reviewer’s suggestion, we have now imaged J8-GECl cells interacting with SLB2s presenting a three-fold increased density of our APC-based glycoprotein glycocalyx. This data is now presented in Supplementary Figures 2k-n. The increased fluorescence of the glycocalyx indicated that the CD45 and CD43 density increased, whilst the densities of other proteins such as CD58 were kept constant (Supplementary Figure 2k). Altering the model glycocalyx in this way had modest, if any effect, on signaling times (Supplementary Figure 2l). Analysing J8-GECl cells on the SLBs, however, we found that they were slower to penetrate the denser glycocalyxes to generate an initial close contact, as expected (Supplementary Figure 2m). After signaling, the cells exhibited reduced spreading and close contact formation, and formed less ‘tight’ contacts (Supplementary Figure 2n). However, the areas of the close contacts remained comparably small, consistent with their initial size being established by the dimensions of microvillar tips, rather than the physical properties of the glycocalyx *per se*.

We have now included a new section of the main manuscript titled “**Impact of glycocalyx density on close-contact formation**” that describes these results (see lines 204-219).

Finally, we have also changed the Discussion to read (see lines 384-387):

“Increasing the glycocalyx density three-fold compromised the efficiency of the T-cell response, consistent with the effects, e.g., of the altered composition of the cancer cell glycocalyx. We note that such effects could be further exaggerated for deeper as well as denser glycocalyxes.”

One related question is whether the two glycocalyx proteins are sialylated, as they apparently should be.

Expressed in human embryonic kidney cells, we would expect our proteins to be heavily sialylated. Confirming this, we placed either no proteins, CD43, or CD45 on an SLB and labelled this with fluorophore-conjugated wheat germ agglutinin (WGA), which binds N-acetyl-D-glucosamine and sialic acid. Our data shows that WGA binds the SLB strongly, and only in the presence of CD43 or CD45 (see Figure i below).

Figure i. 2% Ni-NTA SLBs were made as described in the manuscript. SLBs were incubated with 2.5 ng/uL CD45, 1.5 ng/uL CD43, or PBS for 30 min and washed with PBS. Next, SLBs were incubated with 1 μM WGA-SiR for 15 min and washed with PBS. 3 SLBs were imaged with TIRFM and the

average intensity calculated. The boxplot shows mean \pm SD of 3 FOVs on 1 SLB each. WGA bound specifically to CD45 and CD43 presented on the SLB.

Given the expectation that the proteins would be normally glycosylated, we haven't modified the manuscript in respect of this result.

2. The authors often observe black holes forming on the image plane when cells are interacting with the surface, and interpret these holes as areas depleted of the glycocalyx proteins. However, no proof is provided for such a depletion. The black holes could be, for example, areas where fluorescence is simply quenched. One way to prove protein depletion is to repeat the experiments with the proteins immobilized on a non-SLB (glass?) surface and show that in that case black holes are not forming.

We thank the reviewer for this suggestion. We have now performed this experiment (see **Figure ii** below). The SLB2 protein mix was incubated on glass and imaged using TIRFM. After bleaching an area of the labelled proteins no fluorescence recovery was observed, confirming immobilization. Importantly, no formation of holes under the T cells was observed either, even though the T cells clearly spread on the surface. This data suggests that the formation of 'black holes' is driven by microvillar-driven exclusion of the SLB glycocalyx. Summary data for this result is included as Supplementary Figure 2j.

Figure ii. SLB2 + pMHC^{9V-hi} protein mix was prepared as for making SLBs and incubated on glass coverslips (Piranha and plasma cleaned) for 1h. J8-GECI cells were labelled with CellMask red and added while following the interaction using TIRFM. CD45/CD43 were labelled with AF555 and the images flatfield corrected using the first 10 frames. No distinct close contact formation was observed on glass. Proteins were immobile on glass as fluorescence intensity did not recover after bleaching.

We have now included the following sentences in the main manuscript (see lines 145-148):

“We also confirmed that the holes in the fluorescence were formed by exclusion of the glycocalyx, rather than other photophysical effects, by showing that fluorescent CD43/45 immobilized directly onto a glass surface exhibited uniform fluorescence during cell-contact formation (Supplementary Figure 2j).”

3. It is unclear how CD2 is involved in the processes discussed here- is there evidence that it is enriched on microvilli, or is it distributed over the whole cell body? This should be discussed and potentially experiments to shed light on this issue should be carried out.

Throughout the text we have emphasised the key role of CD2. Supplementary Figure 3h and Figures 4, 5 show that CD2 supports pMHC detection at microvillar contacts. CD2 establishes close contacts that persist for longer (Supplementary Figure 3h), exclude the inhibitory phosphatase CD45 (T-cell surface) over a larger area (Figure 5c), and increase Lck recruitment within close contacts (Figure 5g). According to our explanation for signaling^{1,2}, collectively, this would increase the dwell-time of the TCR/pMHC complex within a phosphatase depleted close contact, increasing the probability of signaling. However, we also note that there is an effect of the cytosolic tail of CD2 on antigen sensitivity (Figure 5f), implicating the recruitment of Lck in increased T-cell signaling.

We have now clarified the role of CD2 in the Discussion (see lines 376-379):

“Each close contact was stabilized by the interaction of CD2 with CD58; removing CD58 reduced the sensitivity of recognition ~100 fold. Sensitivity was enhanced by CD2 via increased close-contact persistence, CD45 exclusion over a larger area, and increased Lck recruitment within close contacts.”

Regarding the distribution of CD2, there is just a single study suggesting that it is enriched on microvillar tips³. Using confocal imaging we find that CD2 is present on the cell body (see **Figure iii** below), but we are yet to determine if we also observe its enrichment at microvillar tips, since relating protein localisation to the 3D topography of cells is a technically challenging task beyond the scope of this manuscript. Regardless of whether it is enriched at the tips of microvilli or not, we observe that CD2 is engaged exclusively at close contacts established by microvilli (Figure 4a in the manuscript).

Figure iii. Confocal images of J8 cells labelled with CellMask Green (to visualize the cell membrane) and CD2-Fab-HaloTag labelled with Alexa-647 dye. Cells were labelled for 1 hour at 4 °C prior before being washed and fixed in 4% paraformaldehyde + 0.25% glutaraldehyde. Fixed cells were placed on PLL-coated surfaces. Images show a maximum intensity projection of 10 frames taken every 200 nm in the z-axis from the midplane of the cells towards the apical plane. This was done to reduce any potential artefacts induced by fixed cells interacting with the PLL-surface at the basal plane. These images clearly show localisation of CD2 (Fab) at the cell body and on protrusions.

4. There is a lot of discussion of the size of features in images. It seems that the diffraction-limited nature of confocal microscopy, which prevents resolving features much smaller than the wavelength, is ignored here, especially when features of less than 0.5 micron are discussed. How does the diffraction limit affect all the size analyses presented in the paper?

We agree with the reviewer that this is an important consideration. Microvilli have a diameter measured by 3D super-resolution in Jurkat T cells of about 214-240 nm⁴. All TIRF imaging was performed with a 1.49 NA objective, which for the emission maximum of the used AF555 dye results in an Abbe diffraction limit of 195 nm (radius of the Airy disk).

We have now addressed sensitivity in Supplementary Figure 6c where we simulated the probability of observing contacts of a given radius and extent of exclusion of the glycocalyx. We find that at 40% exclusion (typically the values we observed for microvillar contacts using TIRF imaging – please see Figure 4d, Figure 4g, Figure 5d, and Figure S6b) we could observe >75% of contacts with a radius of 100 nm (now emphasized in the figure legend of Supplementary Fig. 6c). Our analysis method is therefore sensitive to detecting contacts below the diffraction limit. Contacts smaller than the detection limit would likely be those that result from very transient interactions (< 2 s dwell time, with 2 s being the frequency of our imaging), or the very earliest stage of contacts that we would observe in the next imaging frame. Overall, we do not think our inability to see contacts smaller than 100 nm in radius would affect our conclusions.

Our imaging method is limited by the resolution limit of the optical system. As there is an inherent mutual influence of size and glycocalyx exclusion for small contacts, contacts below and around the diffraction limit will, if detected, appear slightly larger. By choosing an edge-based filter in our close contact analysis (LoG) the effect of exclusion on the detected size was minimised. This correlation is a constant for all the presented data and does not affect the relative change measured. Our image analysis quantifies the contact size based on the number of segmented pixels. A single pixel is therefore the smallest area that can be reported, which is well below the diffraction limit. Temporal and spatial filtering ensures segmented contacts are genuine contacts and not due to noise. We have now clarified these points in the Methods (see Lines 863-876).

Minor points:

1. Page 2- the statement on the time it takes to exclude CD45 molecules is not clear and its context is hard to follow.

This has been changed in the manuscript (see lines 47-51)

“...whereas the impact of CD45 on cell adhesion is less well-studied, it has been estimated that the spontaneous local exclusion of CD45 from a 100 nm diameter region of cell surface, allowing microvillar contact, would take 109 s (31.7 years). This suggests that there will be a requirement for T cells to physically exclude CD45 on opposing surfaces to establish contacts”

2. Page 3- the acronym SLB should be defined the first time it appears.

This has been changed in the manuscript (see line 75). To avoid an abbreviation in a section heading we have defined SLB at the time of next use.

3. Page 4- please define the specificity of “1G4 TCR”.

This has now been addressed in the manuscript (see lines 82-84).

“The complex of the ‘9V’ variant of the cancer/testis NY-ESO-1 peptide (157-165; SLLMWITQV) and HLA-A2 (pMHC9V), which binds the 1G4 TCR47 with a 3D KD of 7.2 μ M39, was used as the agonistic ligand.”

4. Page 5- it is not clear what the dwell-time kinetics discussed here are and how they are quantified.

We have now changed the “dwell-time kinetics” of microvillar tips to their “persistence” to distinguish this metric from the dwell-time of the TCR within close contacts:

Lines 133-135: *“The puncta persisted for ~8 s (Supplementary Fig. 2d); similar observations were made previously for primary T-cells.”*

The persistence comprises the period of time distinct 0.45 μ m diameter membrane puncta (presumably microvilli tips, based on their size) are visible (*i.e.*, observable in the evanescent field of the TIR illumination). Quantification of this is explained in the Methods (see Lines 773-775).

5. Page 6- please provide a general explanation on how close contacts are quantified.

We have added a sentence detailing how contacts were quantified on page 6.
 Line 140-142: *“To analyze protein localisation at these sites, we developed a custom script to segment the images, based on hysteresis thresholding (see Methods for details).”*

6. Same page- the different treatments affect microvilli in different ways, some paralyzing them, some probably collapsing them. These differences and their effects should be clarified.

We have added a descriptor of the drugs and how they relate to altering the actin cytoskeleton and microvilli in the legend of Figure 2a.

Figure 2a legend, lines 1209-1215: *“Several actin-modifying drugs were used to modify T-cell microvilli. Jasplakinolide (Jasplak) enhances the nucleation and stabilisation of actin filaments, resulting in ‘paralysis’ of the actin cytoskeleton and microvillar activity. Latrunculin A (Lat A) and cytochalasin D (CytoD) prevent the formation of actin filaments by sequestering actin monomers and by blocking recruitment of actin monomers to pre-existing filaments, respectively, resulting in filament breakdown and subsequent loss of membrane topography..”*

7. Page 8- It requires more than a single movie to convince that 1-2 pMHC molecules are involved in the contacts observed here. More data and quantification are needed.

We agree with the reviewer and have now repeated this experiment (see **Figure iv**). We did not wish to be too emphatic about the point made here, and only wanted to convey the impression that very few agonistic pMHC are engaged within the first few contacts prior to or at calcium signaling.

Figure iv. a-b Repeated experiment of imaging single pMHC binding in close contact zones. J8-GECl were added to SLB2 presenting pMHC^{9V} (incubated at 1.25 pg/μL) to achieve single-molecule densities. CD45/CD43 was labelled with AF555 and pMHC^{9V} with AF647. The exposure time to image pMHC was set to 200 ms to achieve motion-blur of unbound molecules. Initially 5 frames of the whole FOV were taken without cells for flatfield correction of the CD45/CD43 channel and to estimate the analysis threshold. Several FOVs of cells were taken and manually cropped and sorted into signaling and non-signaling cells. For analysis, a rolling-ball average of 5 frames was calculated and Gaussian blurred with a sigma of 1 pixel. The local maxima above a set threshold were counted to report the number of bound pMHC per timepoint. The timepoint of signaling was determined as the maximum in the calcium signal gradient and cells aligned with it.

a Representative FOVs of a signaling cell before and at calcium signaling. The pMHC, CD45/CD43 signal indicating close contacts (CC), the detected signal peaks, and calcium signal are shown.

b For each cell the maximum number of detected bound pMHC or the maximum number of detected bound pMHC per close contact were calculated and plotted versus the maximum calcium signal. Data comprises 8 FOVs with signaling cells and 5 FOVs with no or non-signaling cells.

c Same cells and SLB as in **a-b**. After taking an image of the CD45/CD43 channel, consecutive frames were taken at 30 ms exposure time to obtain single-molecule photobleaching steps. Four photobleaching time courses are shown. Single step photobleaching (traces 1 and 3) confirmed single pMHC^{9v} molecules were bound within close contacts.

The key data from panels a and b have now been added to the manuscript (Figure 3n, o).

8. Same page- “suggesting that CD2 enhance pMHC detection...”- the way this conclusion is reached is not clear.

This point has now been made more clearly.

Lines 253-256: *“The TCR colocalised in regions of CD58 accumulation (**Movie S13**), suggesting that CD2 would be well placed to enhance pMHC detection at close contacts, by controlling membrane separation at the contact (since CD2/CD58 and TCR/pMHC complexes have similar dimensions.”*

9. Same page- ICAM-1 accumulation in rings- but the features observed don't look like rings at all.... There's a diffuse fluorescent region with a hole in the center, but why call this a ring?

Prior to calcium signaling, we see regions of CD58 accumulation surrounded by 'ring-like' structures of ICAM-1 (Figure S4b). The more diffuse accumulation of ICAM-1, which likely allows better stabilisation of the contact, does not occur until cells have signaled and spread. We therefore think it is fair to refer to these pre-signaling structures as micro-adhesion rings (*i.e.*, a region of CD58 accumulation surrounded by ICAM-1 accumulation). 'Micro-adhesion rings' were first described by Hashimoto-Tane et al.⁵, as areas of TCR accumulation surrounded by integrin engagement.

10. Page 9- “In the absence of CD58 cells were also more mobile”- this requires a better explanation. For example, what the displacement tracks in Figure 4j are is not very clear.

We have now changed “mobile” to “displacement” as we think this better emphasises the effect that we're referring to, and indicated why displacement occurs:

Line 283-284: *“In the absence of CD58 the cells exhibited greater displacement, i.e., the cells ‘drifted’ across the SLB, consistent with CD2 enhancing SLB adhesion generally (**Fig. 4j, k**)”*

Figure 4j legend, lines 1280-1281: *“Displacement is typically due to the currents generated from adding cells into the well of the SLB, which was exploited to probe the adhesiveness of different SLB compositions.”*

11. Page10- the exclusion of CD45 from areas of CD58 accumulation is discussed. In this context, it has recently been shown the CD45 is completely excluded from microvilli tips (Jung et al., <https://doi.org/10.1038/s41467-021-23792-8>). How does this finding reflect on the way the authors discuss here the role of CD45?

This is a controversial matter since Jung et al., 2021 (claiming tip exclusion) contradict the findings of Jung et al., 2016 (claiming uniform/random distribution)^{6,7}. We prefer to be very circumspect about these claims for now. We have modified the manuscript to read:

Line 408-410: *“Finally, whereas Cai et al. did not observe “profound” exclusion of CD45, 30-40% local exclusion of CD45 at close contacts was readily demonstrable in our experiments. We note, however, that it is also proposed that CD45 is “pre-excluded” from the tips of microvilli.”*

12. Page 20: when multiple measurements are taken, is the focus stable enough and doesn't it drift?

We can reassure the reviewer that every effort was taken to prevent drift. For example, to reduce thermal drift, wells were acclimatised to 37°C. We also pre-heated the microscope and objectives to 37°C for 1 hour prior to use, pre-heated metal chambers used to hold the glass slides, and washed

the SLBs using pre-warmed imaging buffer. A representative example of the intensity trace during an FCS recording is shown below for 3x 10 s recordings at a single spot on an SLB (**Figure v**).

Figure v. Representative fluorescence intensity traces taken from one point fluorescence correlation spectroscopy (FCS) experiment on SLB2 presenting pMHC^{null} and Alexa-647 labelled pMHC^{9V}. The FCS intensity trace is for Alexa-647 on the SLB. Each intensity trace was taken sequentially and automatically at the same spot on an SLB for 3x 10 s measurements. As there is no decrease in intensity within each 10 s window and individual traces overlap, this highlights how the system was not experiencing drift.

13. Same page: can equations be written more appropriately, using e.g. subscript and superscript letters when necessary?

This has now been addressed in the manuscript;

Line 583: " $G(\tau) = G(0) \cdot GD(\tau) \cdot GT(\tau) + Off$ "

Line 587: "*confocal volume* ($G(0) \propto N^{-1}$)."

Line 590: " $D = d^2 / [8 \cdot \ln(2) \cdot \tau D]$ "

Reviewer #2 (Remarks to the Author):

In this interesting study, the authors expand the repertoire of tools for studying early T cell:APC engagement, signaling and synapse formation. Thus, they start with the now-standard supported lipid bilayer (SLB), which includes ICAM-1 and peptide-MHC, and add to it CD58 (a ligand for CD2), and CD43 and CD45 (large glycosylated proteins that are a key part of the “glycocalyx”). Using this more physiologically representative SLB (dubbed “SLB2”), they show that the more complex SLB setup allows for a fine T cell response that balances sensitivity and specificity. Interestingly, including only CD58 or CD43/CD45 in the SLB1 setup leads to loss of this balanced response. This led to the authors to further investigate the reasons for the difference in these divergent responses. In brief, they find that the earliest, probing, interactions of T cells and the SLB’s are mediated by microvilli, which can penetrate the glycocalyx. These contacts are further regulated by CD2:CD58 interactions, which limit the size of the contacts. Together, this study provides significant insight into previous observations that demonstrated the presence and dynamic nature of T cell microvilli. It also provides strong support for future use of the SLB2 arrangement as a more biologically relevant model for studying the earliest events in T cell activation.

Overall, this is clearly presented manuscript, including the writing, figures and supplemental movies, the latter of which nicely extend on findings shown in the figures. I have a few comments and suggestions for the authors to address:

Response to reviewer #2

We thank the reviewer for their generous comments on the manuscript.

1. I like the inclusion of the Ca²⁺ signaling reporter, which provides a rapid readout for early signaling. However, I do not think it is fair to use it as a shorthand for all signaling, which the authors seem to imply in some places (although this may not have been their intention). Calcium signaling is quite dynamic in T cells, so early bursts of free Ca²⁺ may not fully represent later signaling events, including by other pathways (e.g. Ras/MAPK).

We agree with the reviewer, of course, that calcium signaling is not necessarily indicative of additional downstream signaling events and we have now altered the manuscript to reflect this.

Line 100: *“Adding the glycocalyx to the SLB1 substantially reduced calcium responses”*

Line 113: *“...modest impact on calcium signaling...”*

Line 122: *“specificity and sensitivity of early T-cell signaling”*

Line 126: *“...glycocalyx substantially suppresses early signaling events...”*

Line 157: *“...microvilli produced substantially lower calcium responses”*

Line 158: *“The treated cells were nevertheless capable of calcium signaling...”*

Lines 202-203: *“...antigen during the early stages of T-cell activation.”*

2. On a related note, have the authors examined whether free Ca²⁺ is necessary for downstream events examined in this system? Thus, would Ca²⁺ chelation still allow the early contacts but prevent spreading? Similarly, if Ca²⁺ were chelated immediately after the initial signaling event, would that prevent the spreading and synapsing?

We have now tried chelating calcium using BAPTA-AM prior to placing cells onto SLB2 presenting pMHC^{9V-hi}. Compared to DMSO control, we did not observe any key differences in cellular spreading (cell footprint) or close contact formation, despite BAPTA blocking increases in intracellular calcium levels.

Figure vi. Response of J8-GECI on SLB2 presenting pMHC^{9V-hi} treated with BAPTA-AM or DMSO control. For labelling, the J8-GECI were washed 1x with RPMI + 1 mM EGTA, and then incubated for 20 min at 37°C with 50 μM BAPTA-AM (or an equivalent amount of DMSO as control) and 5 μg/mL CellMask red in RPMI + 1 mM EGTA. Cells were then washed with PBS + 2 mM MgSO₄ before added to the prepared SLB2 and imaged via TIRFM as described in the main methods. Calcium, the cellular footprint, and number of close contacts are shown relative to the time after close contact. Graphs show mean (± SD) of all cells and data comprise 7-13 cells from 2 SLBs.

3. The authors extend their studies by including some experiments with primary CD8+ T cells. I realize that these experiments have some challenges, but I am not completely convinced that the UCHT1 Fab is a fair comparison in all cases. The authors should discuss this.

We agree with the reviewer that there are, perhaps, important differences between pMHC and UCHT1 Fab (e.g., affinity and protein dimensions). However, the use of membrane-bound UCHT-1 Fab as an agonist is supported by the literature, and it has been previously shown to induce late-stage synapse formation in a large proportion of cells^{8,9}. The fact that we see qualitatively similar effects versus pMHC suggests that, in the context of the present experiments, the differences between pMHC and UCHT1 Fab may not be terribly significant. It would of course be a different matter if we were relying only on the Fab.

To defend the use of UCHT1 Fab, we have added the following statement to the manuscript:
 Line 194-196: *“We also examined close-contact formation for primary CD8⁺ T-cells on SLB2s presenting pMHC^{null} ± ~20 molecules/μm² of an immobilized UCHT-1 Fab/HaloTag construct used as a strong agonist [SLB-bound UCHT-1 Fab has been used previously as a ‘pan-specific’ TCR ligand for human T-cells (citing suitable references).]”*

Reviewer #3 (Remarks to the Author):

T cells use finger-like protrusions called 'microvilli' to scan the surface of antigen-presenting cells. The present paper provides a rationale for such strategy. Microvilli have been implicated in force driven penetration of the glycocalyx and they use relatively 'small' adhesive proteins (primarily CD2, a receptor for CD58 that is also endowed with stimulatory properties) to form close contact zone that are thought to be the site of initial TCR triggering. The authors recently showed that artificial and natural TCR ligands enhance receptor phosphorylation and signaling by 'trapping' TCRs in such phosphatase-depleted close contacts. Here, the authors experimentally demonstrated a hypothesis they published in 2019 (see Ref. 61) and that posited that the small and uniform cellular contacts formed by the microvilli upon penetrating the glycocalyx are essential to equip T cells with the capacity for self vs non-self discrimination by T cells.

The authors developed (1) modified SLBs presenting physiological levels of 'null' pMHC, agonist pMHC (pMHC9V), ICAM-1, CD58, CD43 and CD45, and (2) a human CD4-CD8 $\alpha\beta$ + 1G4 TCR-expressing Jurkat T-cell line, expressing additional LFA-1 molecules to match those of primary CD8+ effector T-cells, and a genetically encoded calcium indicator. 'Second generation' SLB (SLB2) presenting CD58 and a glycocalyx together with pMHC and ICAM-1 allowed the interplay between the glycocalyx and adhesion proteins to be investigated. They demonstrated that the glycocalyx alone substantially suppressed signalling and that together with adhesive proteins it enforced the specificity and sensitivity of antigen recognition by 1G4 T cells. Using labeled CD43 and CD45 molecules inserted onto SLB2 and TIRF microscopy, 'holes' were observed to form in SLB2 corresponding to local exclusion of glycocalyx elements and corresponding to regions of heightened TR density and ZAP-70 fluorescence. These close contacts were reduced upon treatment with actin-modifying drugs. Using a three-color TIRF analysis gauging calcium signalling, interaction footprint, and exclusion of glycocalyx components on the SLB2 presenting pMHCnull and two levels of pMHC9V ligands, four distinct stages of contact were identified and denoted as 'scanning', 'searching', 'spreading', and 'synapsing'. The transition from scanning to searching was independent of pMHC, whereas the progression to spreading was pMHC dependent. Moreover, contacts formed prior to calcium signaling were highly constrained in size. Measurements of contact area at onset of calcium release indicated that pMHC sensing was highly efficient, requiring <0.2% of the total cell surface area. Using single particle tracking on SLBs presenting pMHC9V-lo, 1-2 pMHC9V were found to be engaged at close contacts at onset of calcium release. The TCR was found dispensable for close contact formation and stabilization. In contrast, CD2 enhanced close contact formation and stabilization, whereas ICAM-1/LFA-1 interactions control the cellular footprint that follows calcium signaling. CD2-deficient J8-GECI cell line (CD2KO) were also developed and reconstituted with wild-type CD2 (CD2WT) or CD2 whose cytoplasmic domain was deleted (CD2 Δ CYT). CD2WT and CD2 Δ CYT excluded ~40% and ~32% of CD45 from close contacts, respectively. In marked contrast, CD2KO cells formed fewer and smaller close contacts. Unexpectedly, CD45 exclusion at each CD2KO contact was similar to CD2 Δ CYT.

To examine the effects of increasing contact size on pMHC discrimination, the authors capitalized on J8-GECI T cells that over-express CD2 by a factor of 10 and form 5- to 10-fold larger close contacts than CD2WT-expressing J8-GECI T cells. CD2KO, CD2WT, and CD2WThi cells on pMHC null-presenting SLB2s produced calcium responses of 5%, 20%, and 65%, respectively, revealing a loss of antigenic discrimination as contact size increases. Finally, in contrast to the J8-GECI cells expressing native levels of CD2, approximately 70% of CD2WThi cells produced calcium responses irrespective of pMHC affinity.

The results of this comprehensive, solid, and elegant study that provides a dual 'raison d'être' for the use of microvilli by T cells are fairly discussed.

Response to reviewer #3

We thank this reviewer also for their kind and thoughtful comments on our manuscript.

Minor comments:

1/ The meaning of SLB needs to be specified.

This has been done (see line 75).

2/ Have the authors an explanation for the fact that CD45 exclusion at each CD2KO contact was similar to CD2 Δ CYT?

It seems likely that initial CD45 exclusion on the T-cell side is exclusively a steric effect, resulting from the microvillus contacting the surface, and CD2 might not profoundly influence this. Following contact (and local CD45 exclusion), however, we expect the interaction to be stabilised by CD2 if its ligand is present, as our data suggest. A competing explanation is that CD45 is already partially excluded from the tips of microvilli⁶, although other work suggests that CD45 is homogenously distributed⁷. This requires further investigation.

References

1. Fernandes, R. A. *et al.* A cell topography-based mechanism for ligand discrimination by the T cell receptor. *Proc Natl Acad Sci U S A* **116**, 14002–14010 (2019).
2. Chen, K. Y. *et al.* Trapping or slowing the diffusion of T cell receptors at close contacts initiates T cell signaling. *Proc Natl Acad Sci U S A* **118**, 2024250118 (2021).
3. Ghosh, S., di Bartolo, V., Alon, R., Alcover, A. & Haran, G. ERM-Dependent Assembly of T Cell Receptor Signaling and Co-stimulatory Molecules on Microvilli prior to Activation. *Cell Rep* **30**, 3434–3447 (2020).
4. Sanders, E. W. *et al.* resPAINT: Accelerating Volumetric Super-Resolution Localisation Microscopy by Active Control of Probe Emission**. *Angewandte Chemie International Edition* e202206919 (2022) doi:10.1002/anie.202206919.
5. Hashimoto-Tane, A. *et al.* Micro-adhesion rings surrounding TCR microclusters are essential for T cell activation. *Journal of Experimental Medicine* **213**, 1609–1625 (2016).
6. Jung, Y., Wen, L., Altman, A. & Ley, K. CD45 pre-exclusion from the tips of T cell microvilli prior to antigen recognition. *Nat Commun* **12**, 1–16 (2021).
7. Jung, Y. *et al.* Three-dimensional localization of T-cell receptors in relation to microvilli using a combination of superresolution microscopies. *Proc Natl Acad Sci U S A* **113**, E5916–E5924 (2016).
8. Schubert, A. D. *et al.* Self-reactive human CD4 T cell clones form unusual immunological synapses. *Journal of Experimental Medicine* **209**, 335–352 (2012).
9. Demetriou, P. *et al.* A dynamic CD2-rich compartment at the outer edge of the immunological synapse boosts and integrates signals. *Nat Immunol* **21**, 1232–1243 (2020).

REVIEWERS' COMMENTS

Reviewer #1 (Remarks to the Author):

The authors have nicely addressed my comments (and those of the other two reviewers), and I believe the paper can now be published, though I still have two minor comments:

1. Figure i, demonstrating sialylation, is important in my mind and should be in the supporting information.
2. CD2- I am aware of the paper showing CD2 on microvilli; the question (which should have been phrased better) was regarding the distribution of CD2 on the authors' cells. Their answer is satisfactory. In any case, they should cite in the paper reference 3 of the rebuttal, as it is relevant to their work.

Reviewer #2 (Remarks to the Author):

The authors have sufficiently addressed my previous critiques. I have no additional comments. This is an interesting and innovative study.

Reviewer #3 (Remarks to the Author):

None

“Antigen discrimination by T cells relies on size-constrained microvillar contact”

Jenkins et al.

Response to the reviewers

Reviewer #1

The authors have nicely addressed my comments (and those of the other two reviewers), and I believe the paper can now be published, though I still have two minor comments:

1. Figure i, demonstrating sialylation, is important in my mind and should be in the supporting information.

Figure i has now been incorporated into the manuscript's Supplementary Information as Supplementary Figure 1H.

2. CD2- I am aware of the paper showing CD2 on microvilli; the question (which should have been phrased better) was regarding the distribution of CD2 on the authors' cells. Their answer is satisfactory. In any case, they should cite in the paper reference 3 of the rebuttal, as it is relevant to their work.

Reference 3 (Ghosh et al., 2020) is now cited. We state: “CD2 is reported to be enriched at microvillar tips, which would potentiate CD2 engagement during initial contacts⁵⁵”.

Reviewer #2 (Remarks to the Author):

The authors have sufficiently addressed my previous critiques. I have no additional comments. This is an interesting and innovative study.

Reviewer #3 (Remarks to the Author):

None